# A Secure Data Sharing Model Utilizing Attribute-Based Signcryption in Blockchain Technology

**DOI:** 10.3390/s25010160

**Published:** 2024-12-30

**Authors:** Chaoyue Song, Lifeng Chen, Xuguang Wu, Yu Li

**Affiliations:** College of Cryptography Engineering, Engineering University of PAP, Xi’an 710086, China

**Keywords:** blockchain, attribute-based signcryption, data sharing, privacy security, IPFS, audit

## Abstract

With the rapid development of the Internet of Things (IoT), the scope of personal data sharing has significantly increased, enhancing convenience in daily life and optimizing resource management. However, this also poses challenges related to data privacy breaches and holdership threats. Typically, blockchain technology and cloud storage provide effective solutions. Nevertheless, the centralized storage architecture of traditional cloud servers is susceptible to single points of failure, potentially leading to system outages. To achieve secure data sharing, access control, and verification auditing, we propose a data security sharing scheme based on blockchain technology and attribute-based encryption, applied within the InterPlanetary File System (IPFS). This scheme employs multi-agent systems and attribute-based signcryption algorithms to process data, thereby enhancing privacy protection and verifying data holdership. The encrypted data are then stored in the distributed IPFS, with the returned hash values and access control policies uploaded to smart contracts, facilitating automated fine-grained access control services. Finally, blockchain data auditing is performed to ensure data integrity and accuracy. The results indicate that this scheme is practical and effective compared to existing solutions.

## 1. Introduction

With the rapid development of the internet and advanced technologies, particularly the widespread application of the Internet of Things (IoT), society has now formed an interconnected ecosystem where everything is connected [1]. This situation has led to a dramatic increase in data generation across various fields. Specifically, data generated in the medical field include patient visit records, electronic health records, medical images, and diagnostic reports, which are crucial for disease prediction and medical quality assessment [2]. Meanwhile, data collected by IoT devices encompass images, audio, video, and various digital signals, which facilitate intelligent monitoring and automated control [3]. Educational data include students’ academic performance, teacher evaluations, and competition results [4]. The core of this phenomenon lies in the data-intensive environment brought about by technological advancements, where each type of data not only exhibits a massive volume, but also high diversity and complexity. Simultaneously, this vast amount of data provide rich information resources for analysis, decision-making, and optimization across various industries. However, if these data contain privacy-sensitive information, unauthorized use may lead to severe consequences. Therefore, higher demands are placed on data processing, storage, and analysis capabilities. Specifically, ensuring proper authorization and restricted access to these data is a significant research topic for scholars and a critical challenge faced by current technological and industrial development. Currently, access control strategies are key measures for managing resources and data security, having become standard security features in many system environments. Among them, Attribute-Based Encryption (ABE) has significant advantages in implementing access control [5,6], primarily due to its flexibility in providing access control based on user attributes (such as roles, departments, positions), dynamic permission management, support for complex policies, and enhanced data security. However, traditional ABE technologies primarily focus on data confidentiality and provide less assurance for data integrity and authenticity [7]. To address this shortcoming, the Attribute-Based Signcryption (ABSC) algorithm [8] has emerged, combining ABE and digital signature technologies to not only ensure data confidentiality but also guarantee data integrity and authenticity. Due to its provision of more secure fine-grained access control, ABSC has been widely applied in various scenarios, becoming a promising and practical security solution.

In the context of the increasing prevalence of cloud storage technology, data storage in the cloud not only saves local storage and maintenance costs but also facilitates data retrieval [9]. Additionally, blockchain is another data management technology that provides new solutions for secure, reliable, and efficient data exchange and value transfer [10]. It features decentralization, immutability, and transparency, with data distributed across multiple network nodes. Each executed transaction is recorded in a block, which is linked to the next block through an internal hash pointer, forming a blockchain. This mechanism ensures that once data are written into the blockchain, it cannot be altered arbitrarily, thereby maintaining data consistency and integrity. Consensus algorithms and synchronous network mechanisms also ensure data consistency. However, retrieving data stored on the blockchain are slower and less efficient than traditional databases. Therefore, combining cloud storage and blockchain for data storage has become a new paradigm [11,12]. However, the centralized storage model of cloud servers relies on one or multiple central servers. If these servers fail or go down, data may become inaccessible, affecting business continuity. Additionally, there are data security risks, high costs, and scalability limitations, particularly under high loads or large-scale data transmissions, which may lead to performance bottlenecks.

In response to the centralized storage issues, decentralized storage solutions have gradually become preferred, especially in applications requiring high security, reliability, and flexibility. The InterPlanetary File System (IPFS) [13], as a decentralized file storage and distribution protocol, utilizes core technologies such as Distributed Hash Tables (DHT), content addressing, and peer-to-peer (P2P) networks to provide higher data integrity and consistency. In this architecture, data no longer rely on a single central server but is distributed across multiple nodes in the network, enhancing system reliability. Additionally, IPFS effectively avoids redundant storage through deduplication technology, reducing storage resource usage and optimizing data management and transmission efficiency, ensuring effective utilization of storage space. Wu et al. [14] designed a system model that includes data users, IPFS, and blockchain, proposing the storage of personal health records on IPFS. Although the overall design is innovative and secure, it still presents many security risks and vulnerabilities. First, the lack of a detailed mechanism for fine-grained access control increases the risk of data leakage to unauthorized users. Second, the identity authentication mechanism is not clearly defined, and metadata is stored on the blockchain without encryption, which can lead to privacy breaches and data integrity issues. Azbeg et al. [15] proposed a healthcare data security storage solution based on IPFS and blockchain, where data are encrypted using proxy re-encryption. However, the use of a single smart contract for access control can cause response delays in high-concurrency scenarios. Similarly, Rani et al. [16] introduced a remote patient monitoring scheme based on blockchain utilizing IPFS for data storage and sharing, and DApps for data collection and connection to the blockchain. This solution also faces the limitation of relying on a single smart contract for access control. Jayabalan and Jeyanthi [17] proposed an electronic health record management system combining blockchain and IPFS. Although the framework incorporates multiple security mechanisms, it still faces some risk challenges. Specifically, in a patient-centered access model, there are issues with key management, and if attackers intercept digital envelopes, they may attempt to crack the symmetric keys, leading to data decryption and privacy leaks.

In the field of data sharing, solutions based on attribute-based encryption and blockchain technology have been extensively researched and applied. Although these schemes have made significant progress in enhancing data security, privacy protection, and traceability, they still face several critical challenges and limitations in practical applications. Firstly, many attribute-based encryption schemes rely on complex key management mechanisms, which not only increase system overhead but also introduce single points of failure, posing a threat to overall security. Secondly, while blockchain’s transparency ensures data integrity and trustworthiness, it can sometimes conflict with the need to maintain the confidentiality of sensitive information. Additionally, performance bottlenecks in blockchain, such as low transaction processing speeds and limited data storage capacity, make it difficult to meet the demands of high-concurrency scenarios.

Therefore, to address the aforementioned issues, we propose a novel data sharing scheme based on attribute-based signcryption and blockchain technology. This scheme leverages the advantages of attribute-based signcryption and smart contracts to achieve efficient and secure access control and data sharing. Additionally, it utilizes the immutability and transparency of blockchain to ensure data integrity and verifiability, while integrating IPFS’s decentralized storage to enhance data access efficiency and resistance to censorship. Moreover, by storing the signcrypted ciphertext of original data on IPFS and saving the corresponding hash values on the blockchain, the scheme optimizes resource usage and improves storage efficiency, thereby enhancing data confidentiality.

The primary contributions of our proposed solution are detailed as follows:a.By proposing a dual authentication mechanism based on smart contracts and trusted authorities, and integrating a multi-attribute-based access control strategy, a more flexible, efficient, and verifiable access control approach was jointly developed.b.A data privacy protection mechanism based on attribute-based signcryption was established, simplifying key management, reducing overhead, and enhancing the security of data sharing.c.Our implementation incorporates an auditor authentication framework to oversee and authenticate blockchain transactions, thereby reinforcing the system’s audit capabilities and ensuring data integrity. Furthermore, we benchmarked our solution against comparable Attribute-Based Signcryption (ABSC) systems. The comparative analysis underscores the superior security and reduced computational complexity of our method.

The structure of the following sections is as follows: Section 2 explores the work associated with the original proposal. Section 3 introduces the background knowledge related to the scheme. Section 4 provides a detailed description of the design of the blockchain-based, verifiable, and attribute-based encryption data storage scheme. In Section 5, we theoretically analyze and discuss the correctness and security of the scheme. Section 6 evaluates the practical effectiveness and performance metrics of the proposed scheme, comparing it with similar ones. Finally, we summarize the main findings and contributions of the paper and offer an outlook on future research directions and potential further work.

## 2. Related Works

### 2.1. Attribute-Based Signcryption

Sahai and Waters [18] introduced a novel Attribute-Based Encryption (ABE) scheme that leverages a fuzzy identity encryption approach, treating the identity as a set of descriptive attributes used as a public key. This scheme facilitates encryption by data holders using the recipient’s public key, providing enhanced secure fine-grained access control for data. Consequently, extensive research has been undertaken by many scholars in this area. Zhang et al. [19] first proposed a taxonomy of Attribute-Based Encryption (ABE), identifying Key Policy Attribute-Based Encryption (KP-ABE) and Ciphertext Policy Attribute-Based Encryption (CP-ABE) as two significant variants. Specifically, KP-ABE is primarily utilized in IoT devices [20], while CP-ABE is commonly applied in data encryption, storage, and sharing [21]. Despite its effectiveness in preventing unauthorized access, attribute-based encryption does not adequately address the issues of data tampering or forgery. To address these concerns, Zheng et al. [22] proposed a new cryptographic method called Attribute-Based Signed Cryptography (ABSC), which combines signature and encryption in a single operation, offering non-forgeability, non-repudiation, and confidentiality simultaneously, with significantly lower costs than traditional “signature and encryption” methods. This innovation has spurred in-depth research and exploration. Gagné et al. [23] designed a new signing scheme based on threshold attributes and demonstrated its security within the standard model, introducing a theory that emphasizes satisfactory signing attributes to ensure both message confidentiality and authenticity. Despite the promise of the ABSC scheme, its computational cost remains high, rendering it less suitable for resource-constrained IoT devices. In response, Wang et al. [24] introduced an access tree-based ABSC scheme that integrates ABE and ABS techniques, effectively reducing computational costs while proving its security in the general group model and stochastic prediction model. Following this, Deng et al. [25] presented the Ciphertext Policy Attribute-Based Signature Cipher (CP-OABSC) scheme, which delegates complex encryption computations to a cloud server, allowing users to manage simpler calculations. Yu et al. [26] proposed a lightweight hybrid ABSC scheme that combines Ciphertext Policy Attribute-Based Encryption (CP-ABE) and Key Policy Attribute-Based Signatures (KPABS), ensuring that signatures maintain a constant size and support public verification to tackle the resource limitations faced by IoT devices. Furthermore, this scheme outsources the majority of computational burdens, including signatures, verification, and decryption, to fog nodes. Lastly, Miao et al. [27] employed Attribute-Based Priority Trees to facilitate fine-grained access control with verifiability and searchable capabilities. Rao et al. [28] introduced a secure searchable Attribute-Based Signcryption Scheme, which supports efficient search over signed-and-encrypted data, keyword privacy protection, and self-verification of search results. Despite its contributions, the scheme’s complexity, performance overhead, and security concerns regarding cloud servers cannot be overlooked. For example, the verification process may be overly complex for ordinary users, increasing the likelihood of operational errors. Additionally, computationally intensive tasks are executed locally or on devices with limited resources rather than being outsourced, leading to increased response times and inadaptability to high-concurrency scenarios. Furthermore, cloud servers log users’ search behaviors and results, and any leakage of these logs could compromise user privacy. Zhang et al. [29] presented a comprehensive review of machine learning applications in smart grids, with a particular focus on the security implications of power system characteristics in machine learning-based smart grid applications (MLsgAPPs). This review fills a significant research gap in this area, particularly addressing the issues of security and adversarial attacks in MLsgAPPs, while providing practical application insights and future research directions. However, the discussion lacks detailed exploration of mathematical models or implementation specifics, limiting readers’ understanding of certain technical details. Moreover, the dynamic nature of smart grids necessitates access control mechanisms that can rapidly adapt to changes, yet the access control methods mentioned in the paper fail to update permissions promptly.

While the aforementioned scenarios provide security for data storage, these measures rely on the assumption that cloud servers operate in a “semi-honest and busy” manner. If a single critical cloud model or component fails, the entire cloud server may experience downtime, rendering our data inaccessible and potentially disrupting normal business operations. To more comprehensively address these challenges, we need to consider more diverse and robust data storage and backup strategies, as well as stricter oversight mechanisms for service providers, ensuring that data integrity, availability, and security are upheld across all dimensions.

### 2.2. Blockchain-Based Secure Data Sharinge

The first practical application of blockchain technology is Bitcoin [30], originally designed to ensure the security and transparency of Bitcoin transactions. Blockchain technology achieves data immutability and security through chained structures and cryptographic hashing mechanisms, making it highly effective for managing digital assets and other scenarios requiring high security. With ongoing research, blockchain is now widely utilized across various fields, including the Internet of Things [31], healthcare [32,33], smart homes [34], and cloud computing [35]. Li et al. [36] developed a blockchain-based storage and sharing scheme for educational records, where traditional memories in the chain contain encrypted storage of the original educational data, and the chain merely stores hash values. This approach reduces computational costs and enhances storage efficiency; however, the reliance on a centralized storage server for off-chain data presents a risk of data loss or damage. Wang et al. [37] proposed a blockchain-based access control framework for cloud storage, which combines Ether with attribute-based ciphertext policy encryption to regulate and track data. However, since the cloud storage platform operates under a semi-honest model, the scheme lacks a thorough study of data integrity to ensure that files uploaded by data holders are not tampered with.

The InterPlanetary File System (IPFS) is a decentralized storage protocol that assigns unique hash values based on file content, effectively preventing data duplication and saving storage space, making IPFS more suitable for data storage compared to traditional cloud solutions. Currently, many researchers are exploring the integration of IPFS and blockchain technology to enhance data storage capabilities. Kumar et al. [38] proposed a secure storage solution for healthcare utilizing both IPFS and blockchain. Ullah et al. [39] designed a decentralized storage scheme based on blockchain and IPFS that supports fine-grained access control, where such permissions are defined by attribute-based access control policies. Gao et al. [40] designed a decentralized storage scheme based on blockchain and IPFS. This scheme employs ciphertext-policy attribute-based encryption (CP-ABE) to encrypt symmetric keys, supporting fine-grained access control where the granularity is based on attribute-based access control policies. Zhang et al. [41] developed a chameleon signcryption mechanism and designed a ciphertext authentication protocol, which excels in ensuring identity verification between avatars and the public verifiability of encrypted identities, making it suitable for metaverse scenarios. However, it requires enhancement in the applicability of access control mechanisms and the establishment of comprehensive auditing and monitoring systems to promptly detect and respond to anomalous behaviors, thereby preventing potential attacks. Building on the existing literature, we propose a blockchain and multiple-attribute-based data security storage scheme in which the data holder applies a signed secret algorithm to the data, and the ciphertext is stored on IPFS.

## 3. Preliminary

### 3.1. Bilinear Map and Hardness Assumptions

Bilinear Maps: Define $G_{1}$, $G_{2}$ and $G_{T}$ to be three multiplicative cyclic groups of which the order is p, all of which are prime p. Suppose that $g_{1}$ is a generating element of $G_{1}$ and $g_{2}$ is a generating element of $G_{2}$. A mapping $e:G_{1}\times G_{2}\to G_{T}$ is called a bilinear map [42] if it satisfies the following properties:

(1)Bilinear: $\forall g_{1},g_{2}\in G_{1}$ and $\forall c,d\in G_{1}$, $\;e\left(g_{1}^{c},g_{2}^{d}\right)=e{\left(g_{1},g_{2}\right)}^{cd}$

(2)Non-degeneracy: $\exists R,W\in G_{1}$ fulfills $e\left(R,W\right)\neq 1$

(3)Computability: For $\forall g_{1},g_{2}\in G_{1}$, it is possible to compute $e(g_{1},g_{2})$.

Discrete Logarithm Problem (DLP) [43]: Given a multiplicative cyclic group $G_{1}$ of prime order p with generator g, and given $g,g^{c}\in G_{1}$, it is computationally infeasible to determine c.

Computational Diffie-Hellman Problem (CDHP) [43]: Given a multiplicative cyclic group $G_{1}$ of large prime order p with generator g, and given $g,g^{c},g^{d}\in G_{1}$, it is computationally infeasible to determine $g^{cd}$.

### 3.2. ABAC Model

Attribute-Based Access Control (ABAC) is an access control model that determines whether to grant access permissions by considering various attributes, including those of the subject, object, permissions, and environment [44]. ABAC does not require pre-defined relationships between requesters and data; instead, it makes access decisions by analyzing requester attributes (such as identity, role, and location), object attributes (such as resource type and location), operation attributes (such as read, write, and delete), and environmental attributes (such as time and system status). This model offers fine-grained access control, allowing for dynamic policy adjustments, making it easy to manage and scalable to meet complex access control requirements. Access permissions are determined through the dynamic assessment of the alignment between the attributes of the requester and those of the resource. The core of ABAC lies in defining access policies and decisions based on attributes. This model is described by the quadruple $A\in \left\{S,O,P,\left.E\right\}\right.$, where each field has a specific meaning, detailed as follows.

A means attribute. In the model, attribute is represented as a pair consisting of “key” and “value”. The “key” here is the identifier or name of the attribute, which is used to uniquely identify a specific attribute, while the “value” is the actual content or data of the attribute.

S represents the subject attribute, which defines the characteristics of the requester, such as identity, role, position and credentials. These attributes help determine the basic identity and permission background of the requester.

O denotes an object attribute, which describes the characteristics of the accessed resource, including the identity, location, department, type and data structure of the object. These attributes define the type and location of resources, thus affecting access control decisions.

P stands for permission attribute, which describes what the subject can do to the object. Such as reading, writing, modifying or deleting.

E shows environment attribute, which relates to the context environment at the time of access request, such as time, system state, security level and current access situation. Environmental attributes provide contextual information for access decision, so that the control strategy can be adjusted based on the actual situation.

With comprehensive consideration of these attributes, ABAC model can formulate fine-grained access control policies to ensure that access decisions are made on the basis of considering requesters, resources, operations and their environmental conditions.

In the attribute-based access control, the attribute-based access control request (ABACR) it decides whether to authorize an access or not by combining the subject, object, permission, and environment attributes, which is defined as follows: ABACR={AS∧AO∧AP∧AE}, which denotes a combination of an access request, and this combination consists of four attributes: AS (subject attribute), AO (object attribute), AP (permission attribute), and AE (environment attribute).

Attribute-Based Access Control Policy (ABACP) achieves fine-grained and flexible access control by combining subject, object, permission, and environment attributes using a logical operator, of which the concrete definition is denoted as $ABACP=\left\{AS\wedge or\vee AO\wedge or\vee AP\wedge or\vee \left.AE\right\}\right.$ [45,46], which represents the access rules for a protected resource and ensures that only requests that satisfy all the necessary conditions are granted access.

### 3.3. IPFS

The InterPlanetary File System (IPFS) is a decentralized file storage and sharing protocol [47]. The core idea is to store files within a distributed network rather than relying on a single centralized server. This decentralized design enhances the system’s fault tolerance and robustness while reducing the risk of single points of failure. Additionally, IPFS generates a unique hash value for each file based on its content, serving as an identifier that ensures the integrity and consistency of the files. Furthermore, IPFS incorporates deduplication mechanisms, which minimize redundant data storage and improve storage efficiency. The Distributed Hash Table (DHT) in IPFS aids nodes in locating other nodes that store specific data, significantly enhancing query speed. In summary, IPFS aims to fundamentally transform traditional file storage and sharing methods by providing a more efficient, reliable, and scalable storage solution through decentralization, content addressing, deduplication, and distributed hash tables.

### 3.4. Formalizing the Presented Program Definition

The presented program consists of four algorithms and a one-step auditing operation, which are described below.

(1)System setup: First, the Credible Authority (CA) executes Algorithm GlobalSetup(ω,U) to generate the system public parameter KP; second, the CA runs Algorithm CASetup(ω,U) to generate the master key MSK and the public key PK, where MSK is to be kept secret and KP is public.

(2)Key Generation: The attribute authority (AA) runs the algorithm to generate the corresponding key by entering the master key, public key, and the respective attribute set. For example, to generate a user key, run the algorithm $DataUserKeyGen\left(PK,\;MSK,\;S_{DU}\right)\;\to \;\left(SK_{DU}\right)$. The scheme utilizes smart contracts to manage and share encrypted transaction information on the blockchain (see Figure 1), which ensures the security and consistency of the information, and also automates the process of verification and execution of transactions.

(3)Signcryption: The data holder generates a signed ciphertext SCt using a Signcrypt algorithm by inputting his private key $SK_{DH}$, message M, and access policy T.

(4)De-Signcryption: The data user enters a private key $SK_{DU}$, an encrypted ciphertext SCt, and an attribute set S to execute the algorithm, and outputs the message M.

(5)Audit: Auditors validate transaction data in terms of both data integrity and timeliness to ensure that there is no tampered or falsified data, helping enhance the trust and reliability of the blockchain by maintaining transparency and ensuring that network rules are enforced.

## 4. Proposed Scheme Design

This section describes the architecture of the proposed model, transactions on the blockchain, the generation of smart contracts, and the definition of the security model.

### 4.1. System Architecture

The structure of the proposed blockchain-based attribute-based secure data IPFS storage scheme is shown in Figure 2 and consists of seven entities, which are data holder, data user, IPFS, trusted authority, attribute authority, blockchain, and auditor. Among them, the authority is composed of trusted agency and attribute agency.

Credible Authority (CA): This mainly generates public parameters and master-private keys needed by data holders and data users. It is assumed that CA is a completely trustworthy entity that will strictly comply with the operation without tampering with data and will not sell user information privately. However, other entities cannot fully trust CA, therefore, on one hand, CA can be added to the blockchain node, which is convenient to improve the efficiency of data processing and the tracking of key leakage; on the other hand, the two entities of identity organization and attribute organization are added.

Date Holder (DH): The data holder signs and encrypts the data using a signed cipher algorithm, and then uploads it to IPFS, which returns a content-based hash address. Then, the DH formulates an access control policy, where users whose attribute conditions meet the requirements can access the data, and encrypts the hash value and symmetric key based on the access control policy, and then uploads the generated ciphertext and the access control policy to the created smart contract, and deploys it to the blockchain.

Data User (DU): DU can access the required data and should fulfill more attributes of the ciphertext access policy.

Attribute Authority (AA): The Attribute Authority is responsible for collecting and storing the attribute information submitted by the user, this entity is honest but curious and will not access unauthorized information.

Interplanetary File System (IPFS): It is mainly used to store ciphertexts uploaded by data holders and also to verify the validity of ciphertexts. If the ciphertext is not valid, IPFS cannot store the ciphertext. Assuming that IPFS is honest, it will operate as required, but there is curiosity to try to decrypt the ciphertext to get the ciphertext data information.

Blockchain: Blockchain captures the access requests and access activities of different users in the form of transactions while storing the smart contracts created by the data holders. In other words, by using blockchain, all transactions are replicated by different node records on the blockchain network. At the same time, it is retrospective in the sense that transactions involving the data holder, any data user or device can be traced back to the data flow and their behavior cannot be denied.

Auditor: Auditors are primarily responsible for verifying and confirming the validity and legitimacy of data and transactions on the blockchain network. It usually includes performing tasks such as verifying transactions, confirming blocks, preventing double-flush payments, and participating in consensus algorithms, and can provide auditing and supervision when necessary.

The scheme is shown in Figure 2, and the specific operation details are described as follows.(1)CA executes the algorithm to generate the master key MSK and sends it to the attribute organization.(2)The attribute organization generates the attribute private keys of the data holder and the data user, where DH receives the attribute private key $sk_{h}$, and DU receives the attribute private key $sk_{u}$.(3)DH creates a smart contract on the blockchain and then encrypts and uploads the formulated access policy to the smart contract.(4)DH formulates encryption policy $\left(t,D_{e},i\right)$ and signature policy $\left(t,D_{s},i\right)$, and then DH signs and encrypts the data through the access control policy, and uploads the signed ciphertext SCt to IPFS.(5)IPFS returns a hash value to DH.(6)DH creates a transaction containing hash, timestamp, signature, etc., and finally encrypts and uploads the transaction to the blockchain.(7)DH generates a hash index to upload to the smart contract.(8)DU requests access to the data on IPFS, first of all, it needs to visits the hash index of smart contract. If the DU attributes meet the requirements of the access control policy, then the DU submits the ciphertext index $index\left(SCt\right)$ to IPFS.(9)If the authentication passes, IPFS returns part of the ciphertext SCt related to user privileges to DU for data recovery.(10)DU decrypts the ciphertext SCt with the decryption key and gets the required IPFS data.(11)The auditor verifies the transactions on the blockchain, including transaction information, timestamps, participant information, and so on.

### 4.2. The Structure of the Blockchain

The blockchain is the core component of a decentralized storage solution, where data storage is dispersed across multiple nodes, and smart contracts can be programmed to manage and enforce access rights, data validation, and sharing rules for the stored data, enhancing the flexibility and security of the decentralized storage solution. The record of data changes on the blockchain is open and transparent, and can be viewed and verified by anyone. The structure of the blockchain is chained and consists of blocks, each containing transaction data and associated metadata. The structure of the block and the design of the smart contract are described specifically below.

#### 4.2.1. On-ChainTransactions

Figure 3 shows that a block contains two elements, namely a block header and a block body. A block header is a central part of every blockchain, which encapsulates information such as timestamp, version number, previous block’s hash value, Merkle Root, etc. A block contains the list of deals, status change, signature of the creator, etc. Taking a data transaction uploaded by a data holder, as an example, the block body contains $Hash_{T}$, $H_{1}\left(index\left(SCt\right)\right)$, and $sign\left(Ts\right)$, where $Hash_{T}$ represents the hash value of the block added to the blockchain at time T, $H_{1}\left(index\left(SCt\right)\right)$ represents the index value of the hash of the signed ciphertext. The signature of the currently generated block trading is indicated by $sign\left(Ts\right)$. Transaction data are stored by the data holder in the blockchain to make it complete and traceable.

#### 4.2.2. Smart Contract

A smart contract is the automatic execution of predefined code and rules without the intervention of a third party when specific conditions are met [48]. And once a smart contract is deployed on the blockchain, it is almost impossible to be changed or deleted because of the unique chain structure, where the hash value of the previous block is linked to the next block, which guarantees the tampering of the transaction, in addition, the result of the operation of the smart contract needs the validation and agreement of multiple nodes of the blockchain network before it can be written on the blockchain, which ensures the uniqueness and reliability of the transaction and the execution of the contract, establishes trust that prevents double payments and fraud. Ether is applied in our proposed scheme because the blockchain is public and can be viewed by all, so the contract and transaction data stored to the blockchain are signed and sealed.

In addition, the proposed scheme enforces the data access and usage rules specified in the protocol through smart contracts, which can provide a solid foundation and guarantee for data exchange between data holders and data users, and ensure that the participants exchange data securely [49]. Figure 1 shows the process by which the data holder makes a smart contract and then gets on the same page as two data users. Next, we will explain the working mechanism of this process in detail.

The data holder creates a smart contract. First creates a smart contract and sets the conditions for its execution, e.g., who can be allowed to access his data.

Smart contract encryption and broadcasting. Encrypt the smart contract using the visitor’s key, followed by broadcasting it across the blockchain network.

Visitor requests access. The smart contract is triggered when user 1 wants to access the medical information.

Verifying the identity of the requester. The smart contract verifies the identity and permissions of data user 1. If data user 1 is not eligible, the access fails.

Access authorization. If data user 1 is eligible, the smart contract allows data user 1 to decrypt the contract and access the information.

Message validation and routing. Data user 1 can view the information and encrypt the contract using data user 2’s key as specified in the smart contract, and then broadcast it again to the blockchain network.

Subsequent Verification. After receiving the contract, data user 2 decrypts the contract using the data holder’s key to verify the accuracy of the information. The data holder can check the content of the contract and return the validation results to data user 2.

### 4.3. Security Model

We proposed scheme meets the demands of confidentiality, unforgeability, verifiability, and privacy. Next, we describe the four security models of the scheme in terms of these four aspects.

#### 4.3.1. Message Confidentiality

The security model is usually defined as being indistinguishable from a chosen access policy in the face of an irresistible chosen-ciphertext attack. Scenario is proved to be indistinguishable from IND-CCA [50] if no attacker A is able to defeat challenger C in the interactive security game competition with a significant advantage in probabilistic polynomial time. The detailed definitions of the attacker A and the challenger C in the interactive game are given below.

(1)Initialization:

The adversary A elects attributes ${S_{A}}^{*}$ on which C is able to base the verification of the security properties during the subsequent interrogation phase.

(2)Setup:

The Challenger C executes the algorithm to generate the public parameter KP and the master key MSK, and sends KP to the adversary A.

(3)Query phase 1:

The A requests a bounded number of queries from C in an adaptive manner.

• Secret Key query:

At this phase of the inquiry, A requests private key SK from C. C invokes the KeyGen algorithm that generates the equivalent SK according to ${S_{A}}^{*}$ submitted by A, and then sends it to A.

• Signcrypt query:

The adversary A selects a specific attribute set ${S_{A}}^{*}$ according to the need, ${S_{A}}^{*}$ satisfies the encryption and signature policy, C calls the KeyGen algorithm again according to ${S_{A}}^{*}$ selected by A and generates SK corresponding to the attribute set ${S_{A}}^{*}$. Then C executes the sign-secret algorithm to process the message M. Firstly, C signs the message M with the private key SK, and then encrypts the signed message using the private key SK and generates the sign-secret message SCt to be sent to A. The sign-secret message SCt is sent to A. Then, C encrypts the signed message with private key SK and sends the signed secret message SCt to A.

• De-signcrypt query:

A submits signed ciphertext SCt and corresponding ${S_{A}}^{*}$ to C. Then, C calls the KeyGen algorithm based on ${S_{A}}^{*}$, and generates the corresponding private key SK. Subsequently, SK is forwarded to A. C decrypts the ciphertext SCt with the generated private key SK to obtain the original message m. After the decryption is completed, A accepts the export effect of C.

(4)Challenge:

An adversary first chooses two messages $M_{0}$ and $M_{1}$ to challenge, and the two messages are of equal length, which is equivalent to $\left|M_{0}\right|=\left|M_{1}\right|$. Moreover, A formulates an access policy T such that the sets $S_{1},S_{2},\cdots ,S_{q}$ satisfy the access policy. And by flipping a coin, the challenger C randomly chooses a bit $b\in \left\{0,\;1\right\}$. C determines a set of attributes S*, executes the KeyGen algorithm, and generates the corresponding private key SK, and then C executes the Sign-Secret algorithm using the private key SK, and generates the Sign-Secret ciphertext $SCt^{*}$, which is provided to A as the challenge ciphertext.

(5)Query Phase 2:

The process is similar to Phase 1, where A may initiate an enquiry for any ciphertext except the ciphertext being questioned.

(6)Guess:

An attacker gives a guess about b, and if b′=b, then A wins the game. Thus, $Adv=|\mathrm{Pr}[b^{\prime }=b]-1/2|$ is denoted as the advantage of attacker A in the game.

(7)Definition 1:

If the probability $Adv_{\left(A\right)}^{IND-CCA}$ of winning the game is negligible for any of the adversaries, then our scheme has the secrecy property under an irresistible chosen-ciphertext attack.

#### 4.3.2. Ciphertext Unforgeability

If A doesn’t prevail in polynomial time the superiority is inescapable. Therefore, this security model is defined as resisting the existence of unforgeability of adaptive choice message attacks under the Choice Attribute Set model. The definitions of A and C are given on the game below.

The initialization and setup process for unforgeability should be consistent with the process described for confidentiality mentioned earlier.

(1)Query Phase 1:

A may originate the following queries at each stage.

• Secret Key query:

The challenger C executes the KeyGen algorithm based on the set of attributes ${S_{A}}^{*}$ provided in the request to generate the corresponding private keys SK and allows A to use these keys.

• Signcrypt query:

For each message $M^{*}$ sent by A, which satisfies the attribute set category, C uses the SecretGen algorithm to generate the private key SK. C then signs the message $M^{*}$ and generates the signed secret ciphertext $SCt^{*}$ to be sent to A.

• De-signcrypt query:

A Submit signed ciphertext $SCt^{*}$ and ${S_{A}}^{*}$. Initially, the SecretGen algorithm is invoked to obtain the corresponding private key SK. Afterward, C executes the Designcryption algorithm followed by sending the decrypted message to A.

(2)Forgery:

In this phase, the attacker A forges a user’s private key sk and proceeds to obtain the ciphertext SCt′ and the corresponding set of attributes $S_{A}^{*\prime }$ of message $M^{*}$ (which A did not obtain through the questioning phase). At this point, C executes the Designcryption algorithm, and if the output result $\left(SCt\prime ,sk,S_{A}^{{*}^{\prime }}\right)=M^{*}\neq ⊥,$ $\left(MSK,sk^{*}\right)\neq ⊥,\left(MSK,sk^{*}\right)\to ID^{*},ID^{*}\notin \left\{{\mathrm{ID}}_{i}\right\}$, where S represents the set of legitimate users in the system, these mean that $sk^{*}$ is punctual and $M^{*}$ No one’s ever heard of it before. It also indicates that A forges successfully and wins the game. $A^{\prime }s$ advantage in this game of interaction is expressed as $Adv_{A}^{Unforgeability}=|\mathrm{Pr}[b^{\prime }=b]-1/2|$.

(3)Definition 2:

The odds of either attacker winning a contest are slim to none, thus showing that the scheme proposed in this paper is existential unforgeability against adaptive chosen message attack.

#### 4.3.3. Verifiability

In this model, adversary A tries to crack the signcryption message, while challenger C takes measures to ensure that even if the signcryption message is leaked, the signer’s identity will not be revealed. Let us introduce the game between A and C.

(1)Setup:

A creates an access control policy T as the public key PK of the query policy request. C runs the algorithm to generate the master key MSK and the public parameter KP, and then forwards the public parameter KP to C. In addition, C continues to perform the algorithm and dispatches the generated PK as a message to A.

(2)Challenge:

The inquiry of attacker A about the private key of attribute set $S_{A}^{*}\left(S_{A1},S_{A2},\cdots ,S_{Aq}\right)$ means that A wants to obtain the private information or key associated with these attributes. The challenger C executes the key generation algorithm to generate the private key SKA, and then returns it to A. Then A requests to signcrypt a certain challenge message $M^{*}$ that meets the attribute set, C operates signcryption algorithm to generate signcryption ciphertext $SCt^{*}$ in response to A, and then delivers $SCt^{*}$ to A. Finally, C decrypts signcryption ciphertext $SCt^{*}$ and conveys $M^{*}$ to A.

(3)Forgery:

The adversary A proposes an access policy T, and T needs to meet the expected access policy of challenger C. This means that the opponent’s strategy is legal and can get the corresponding permissions in the challenger’s system. C uses the access policies provided by messages $M^{*}$ and A to generate a signcryption ciphertext $SCt^{*}$. Then, c attempts to decrypt the ciphertext $SCt^{*}$ to recover the message m. If the recovered message $M^{*}$ is the valid plaintext of the ciphertext $SCt^{*}$, that is, the original message corresponding to the ciphertext $SCt^{*}$, the opponent successfully wins the game.

(4)Definition 3:

If the scheme ensures that any user with the attribute set $S_{DU}$ (provided that the attribute set conforms to the access control policy T) can verify the signature, ensure that the message has not been tampered with and indeed comes from a legitimate sender, and can successfully recover the original content of the encrypted message, then the scheme is considered as “verifiable”.

#### 4.3.4. Privacy

If adversary A can hardly distinguish the sources of two correct signatures generated by the same attribute in this secure game, the identity privacy of the signer is protected, which also means that the scheme maintains a high degree of privacy in calculation. The security game for both A and C is described as follows.

(1)Setup:

A selects the attribute field U and transmits to C, and then C performs the setting steps to generate the common parameter KP and sends it to A.

(2)Challenge:

The opponent A picks the access policy T, two attribute sets $S_{A1}$ and $S_{A2}$ ($\left|S_{A1}\right.\cap \left.S\right|=\left|S_{A1}\cap \right.\left.S\right|=t$) which meet the access policy, and a message M to challenge C. C randomly selects a bit b: $b\in \left\{0,\;1\left.\right\}\right.$, executes the key generation algorithm to generate the key $SK_{C}$, and then executes the signcryption algorithm to generate $SCt_{b}$. Through this process, challenger C tests whether opponent A can effectively crack the signcryption scheme, thus verifying the security of the scheme. The task of opponent A is to process the challenge message $SCt_{b}$ and try to extract the decryption information from it.

(3)Guess:

In case the opponent A′s final output result is b′, and b’=b exists, then A gains the game. The winning advantage of opponent A in the game can be expressed as $Adv_{A}^{privacy}=|\mathrm{Pr}[b^{\prime }=b]-1/2|$.

(4)Definition 4:

If the likelihood $Adv\left[{Exp}^{privacy}\right.\left.\right]$ of winning a race is negligible for any of the adversaries, then our scheme has the secrecy property under an irresistible chosen-ciphertext attack.

### 4.4. Formatting of Mathematical Components

In this section, we give the exact algorithm for the scheme, which covers five main phases, and the specific constructions are described as follows. In order to facilitate the understanding of the analytical model, the symbols used in the model are defined below, as shown in Table 1.

#### 4.4.1. System Setup

In this stage, after data holders, data users, and auditors subscribe to information, the system glues public parameters and master keys.

(1)

GlobalSetup(ω,U)→KP



It performs the algorithm by CA with input attribute domain U and security parameter ω. The output result is public parameter KP. It chooses cyclic groups $G_{1}$ and $G_{2}$ with corresponding generators $g_{1}$ and $g_{2}$, both are elementary p, and there is a bilinear mapping $e:G_{1}\times G_{2}\to G_{T}$. And randomly select two random oracle hash functions: $H_{0}:{\left\{0,\left.1\right\}\right.}^{*}\to Z_{p}$, $H_{1}:{\left\{0,\left.1\right\}\right.}^{*}\to G_{1}$. Next, it has the system public parameter $KP=\left(e,p,g_{1},g_{2},G_{1},G_{2},G_{T},H\right)$.

(2)

$CASetup(\omega ,U)\to \left(PK,MSK\right)$



After the data holder, data user, and auditor registration information have been added to the system, meanwhile, each data holder creates an account in Ethernet and then releases it to others. After registration, CA executes the algorithm which allows security parameter ω and attribute field U as the inputs and the outputs are the generated public key PK and master secret key MSK. A specific encoding function $\nu :U\to Z_{p}$, where U is an attribute domain containing a set of attributes, and $\left|\left.U\right|\right.=q$, q denotes the number of attributes. Then, pick a set $F=\left\{f_{1},\right.f_{2},\cdots ,\left.f_{j-1}\right\}$, in which j+1≤q, pairs of distinct factors in ${\left(Z/pZ\right)}^{*}$. In addition, τ,β are randomly selected from $Z_{p}^{*}$ and calculate η,λ.
(1)$$\eta =g^{\tau \beta }$$
(2)$$\lambda =e\left(g_{1}^{\tau },g_{2}\right)$$

Finally, the public key is as follows:(3)$$PK=\nu \{{G_{1},G_{2},G_{T},e,\{g_{2}^{\tau \beta^{j}}\}}_{\left\{j=0,\ldots ,2q-1\right\}},\eta ,F,\nu ,\lambda ,H\}.$$

Moreover, the master key is represented as follows:(4)$$MSK=\left\{\tau ,\left.\beta \right\}\right.$$

#### 4.4.2. Key Generation

During the current period, AA generates keys for data user and data holder based on their respective attributes, as described below.

(1)

$DataUserKeyGen\left(PK,\;MSK,\;S_{DU}\right)\;\to \;\left(SK_{DU}\right)$



Attribute Authority $AAKeyGen(PK,MSK)\to SK_{U}$: In this algorithm, input the public parameter KP, the master key MSK. For an arbitrary subset $S_{j}\subset U$, a number r is selected at random to meets $r\in {\left(Z/pZ\right)}^{*}$, so the private key of the user $SK_{DU}=({\left\{g_{1}^{\frac{r}{\beta +\nu \left(a\right)}}\right\}}_{a\in S},{\left\{g_{2}^{r\beta^{i}}\right\}}_{j=0,\cdots ,q-2},g_{2}^{\frac{r-1}{\beta }})$.

(2)

$DataOwnerKeyGen\left(PK,\;MSK,\;S_{DH}\right)\;\to \;\left(SK_{DH}\right)$



The data holder utilizes its own attribute generation algorithm by inputting the public parameters, public key, and attribute set of the data holder, and its process of generating the private key is similar to that of the data user attribute private key generation, where the output result is $SK_{DH}$ that then gets sent to the data holder.

#### 4.4.3. Signcryption

The data holder runs the algorithm to sign and encrypt the message M through encryption policy $\left(t,B_{e,i}\right)$ and signature policy $\left(t,B_{s,i}\right)$.Where, $\left(t,B_{e,i}\right)$ refers to the encryption policy, which needs to satisfy the conditions $B_{e,i}\subset U$, $s=\left|\left.B_{s,i}\right|\right.$ and $1\leq t\leq \left|\left.B_{e,i}\right|\right.$. Meanwhile, $\left(t,B_{s,i}\right)$ stands for the signature policy and $B_{s,i}\subset U$, $s=\left|\left.B_{s,i}\right|\right.$ and $1\leq t\leq \left|\left.B_{s,i}\right|\right.$ are its preconditions. The data holder is responsible for constructing the signed ciphertext, the details of which are shown below.

(1)The data holder’s key $SK_{DH}$ and the aggregation function are applied to compute $C_{1}$. The formula for $C_{1}$ is shown below.
(5)$$C_{1}=Aggreg({\left\{g_{1}^{\frac{rDH_{id}}{\beta +\nu \left(a\right)}},\nu \left(a\right)\right\}}_{a\in Hs_{DH}})=g_{1}^{\frac{rDH}{{\prod_{a\in {Hs}_{DH}}}^{\left(\beta +\nu \left(a\right)\right)}}}$$

Then the polynomial $P_{\left(DH_{id},s\right)}\left(\beta \right)$ is computed according to the algorithm.
(6)$$P_{\left(DH_{id},s\right)}\left(\beta \right)=\frac{1}{\beta }\left(\prod_{a\in S\cup D_{n+1-1\backslash DHid}}\left(\beta +\nu \left(a\right)\right)-\prod_{a\in S\cup D_{n+1-1\backslash DHid}}\nu \left(a\right)\right)$$

And because of the presence of $\psi_{1}$, in which
(7)$$\psi_{1}=\prod_{a\in S\cup D_{n+1-1\backslash DHid}}\nu \left(a\right)$$

Thus, the formula for $P_{\left(DH_{id},s\right)}\left(\beta \right)$ can be further expressed as follows:(8)$$P_{\left(DH_{id},s\right)}\left(\beta \right)=\frac{1}{\beta }\left(\prod_{a\in S\cup D_{n+1-1\backslash DHid}}\left(\beta +\nu \left(a\right)-\psi_{1}\right)\right)$$

In the meanwhile, using the key $SK_{DH2}$ of data holder 2, data holder 1 can deduce that $\psi_{2}=g_{2}^{rD,iP_{\left(DH_{id},s\right)}\left(\beta \right)}/\psi_{1}$.

(2)Calculate the signature value. Taking message or data (m) as an example, the signature value of message $M_{1}$ is calculated next, and the specific calculation process is as follows.
(9)$$\phi_{1,1}=C_{1}\cdot g_{1}^{\frac{H\left(M1\right)}{{\prod_{ai\in AA\cap A_{DH}}}^{\left(\beta +\nu \left(a\right)\right)}}}$$
(10)$$\phi_{1.2}=sk_{D3}\cdot \psi_{2}\cdot g_{2}^{H\left(M1\right)P_{\left(DH_{id},s\right)}\left(\beta \right)/\psi_{1}}$$
(11)$$\phi_{1.3}=g_{2}^{\tau H\left(M1\right)}$$
(12)$$\phi_{1}=\left(\phi_{1.1},\phi_{1.2},\phi_{1.3}\right)$$

Then, select a random value from $\varepsilon \in {\left(Z/pZ\right)}^{*}$, and calculate the signcryption ciphertext $\sigma_{1}$, and a specific calculation process is described below.
(13)$$\sigma_{1,1}=g_{1}^{\left(\tau \cdot \beta \right)-\varepsilon }$$
(14)$$\sigma_{1.2}=g_{2}^{\varepsilon \cdot \tau \cdot \prod_{a\in S}\left(\beta +\nu \left(a\right)\right)}$$
(15)$$\sigma_{1,3}=e{\left(g_{1},g_{2}\right)}^{\tau \cdot \varepsilon }e{\left(g_{1},g_{2}\right)}^{\tau \cdot H\left(m1\right)}\cdot M_{1}=\varepsilon \cdot M_{1}$$

So, the generated signed ciphertext is that
(16)$$\sigma_{1}=\left(\sigma_{1,1},\sigma_{1,2},\sigma_{1,3}\right)$$

Therefore,
(17)$$SCt_{1}=\left(\phi_{1.1},\phi_{1.2},\phi_{1.3},\sigma_{1,1},\sigma_{1,2},\sigma_{1,3},P_{\left(DH_{id},s\right)}\left(\beta \right),\psi_{2}\right)$$

(3)

$\mathrm{DH}.\mathrm{CiphertextIndexingGen}\left(\phi_{1},SCt_{1},T\right)\to {\mathrm{EncscT}}_{\mathrm{List}}\;$



According to the algorithm, the data holder 1 takes the access policy T, the signature $\phi_{1}$ of the message m and the signcryption ciphertext SCt of the message as inputs, and the output result is the index $\mathrm{Inx}(SCt_{1})\;$ of the signcryption SCt. Then the hash value $BlockHash_{t1}$ corresponding to the latest block is extracted against the current time $t_{1}$, and $DH_{1}$ creates a new transaction $Ts\left(DH_{1}\right)$. Lastly, let us calculate the transaction data value $BlockHash_{t1}∥H_{1}\left(Inx\left(SCt_{1}\right)\right)$.

(4)Verification and storage

As signed ciphertext $SCt_{1}$ is added to IPFS, IPFS automatically calculates a unique hash value for the uploaded data using content addressing. This hash value is calculated based on the content of the data, and any changes to the data will result in the generation of a different hash value. In order to verify the consistency of the data and ensure that the data uploaded to IPFS are consistent with the records on the chain, then it is very necessary to verify that the hash value $Blockhash_{T}$ computed by IPFS matches the hash value $Hash\left(T\right)$ extracted from the blockchain. Data can only be considered consistent if $Blockhash_{T}$ and $Hash\left(T\right)$ are equal, which means that the hash $Blockhash_{T}$ recorded on the blockchain and the hash $Hash\left(T\right)$ computed by IPFS are hashes of the same block of data. As an example, if $Hash\left(t1\right)$ is a hash value generated at the moment t1, then $Blockhash_{T}$ should be a hash value for the data at the same point in time.

And verify that it matches the equation:(18)$$\lambda =e\left(u_{i}^{-1},\phi_{1,2}\right)\cdot e{\left(g_{1}^{\tau },g_{2}\right)}^{H(m1)\left(1-X_{1}-\frac{1}{X_{1}}\right)}\cdot e\left({\varphi_{1,1}}^{\psi },g_{2}^{\prod_{a\in S\cup D_{n+1-1\backslash DHid}\beta +\nu \left(a\right)}}\right)$$
where $X_{1}=\prod_{a\in S\cup D_{n+1-1\backslash DHid}}\frac{\beta +\nu \left(a\right)}{\nu \left(a\right)}$, if the hash verification is inconsistent, the data are not stored; otherwise, IPFS stores the data and then returns the hash value $Hash\left(SCt_{1}\right)$ of the signed ciphertext to the data holder $DH_{1}$. Finally, $DH_{1}$ generates $\mathrm{Inx}\left(Hash\left(SCt_{1}\right)\right)\;$ and writes it to the smart contract.

#### 4.4.4. De-Signcryption

(1)Data retrieval:

• Access control check:

Users browse smart contracts and view their own attributes and permissions. Because the smart contract will automatically verify whether the access control policy is met according to the user’s attributes and permissions, if the conditions are met, the smart contract allows the user to proceed to the next operation.

• Data download:

Qualified data users can obtain the index hash of the data they need from the smart contract. Then, the hash value is used to retrieve the data file in the IPFS network and download the data file from the IPFS network.

(2)Data verification and decryption:

• Data integrity verification:

Before decryption, users need to ensure that the downloaded data files have not been tampered with. The operation steps are as follows: first of all, the hash value is recalculated. Users use the same hash algorithm to calculate the hash value of downloaded data files. Next, the hash values are compared. The calculated hash value is compared with the hash value stored in the smart contract. If the two hash values are consistent, it means that the data have not been tampered with or damaged during transmission.


(19)
$${\mathrm{Message}\;\mathrm{Integrity}\;=\;\mathrm{Hash}(M)}_{\mathrm{DU}}\;=\;\mathrm{Hash}(M)$$


• Data decryption:

A de-signcryption function process based on consists of the following steps. Data decryption is the key step to obtain the original data. The process of decryption can be different according to different encryption methods. At first, it is necessary to verify that the ciphertext of the randomized value is a signature issued by the DH, at which point the function expression is
(20)$$Sl=e\left(x^{-1},\phi_{2}\right)\cdot e\left(\phi_{1}^{1/U_{1}},g_{2}^{\tau \prod_{a\in S\cup D_{n+1-1\backslash DHid}}\left(\beta +\nu \left(a\right)\right)}\right)\cdot e{\left(g_{1}^{\tau },g_{2}\right)}^{H\left(Rc\right)}=e\left(g_{1}^{\tau },g_{2}\right)$$

Then, in the decryption operation, the user’s attribute information (e.g., department, position, etc.) is the basis for determining whether the user has the privilege to decrypt the ciphertext or not. If the user has multiple attributes, we need to aggregate these attributes to determine whether they comply with the access policy specified in the ciphertext, and the algorithm is as follows. For symmetric encryption (AES), the user directly uses the key to decrypt.
(21)$$C_{2}=Aggreg\left({\left\{g_{1}^{\frac{rDU_{id}}{\beta +\nu \left(a\right)}},\nu \left(a\right)\right\}}_{a\in Hs_{DU}}\right)=g_{1}^{\frac{rDU}{\prod a\in {{Hs}_{DU}}^{\left(\beta +\nu \left(a\right)\right)}}}$$

Building on the previous basis, we compute the aggregated user attribute key $C_{2}$ in combination with the encrypted data ${\left\{h^{r\beta^{j}}\right\}}_{j=0,\ldots ,q-2}$ from the ciphertext of the user’s private key $SK_{DU}$:(22)$${\left[e\left(\sigma_{1,1},\;g_{2}^{rDU_{\left(HS_{DH},S\right)\left(\beta \right)}^{P}}\right)\cdot e\left(C_{2},\sigma_{1,2}\right)\right]}^{\frac{1}{X_{1}}}=e{\left(g_{1},g_{2}\right)}^{\left(\varepsilon \cdot \tau \right)rEU}$$

Hence, the decryption key is generated according to the algorithm as:(23)$$dk=e{\left(g_{1},g_{2}\right)}^{\tau \cdot \varepsilon }\cdot e{\left(g1,g2\right)}^{\tau \cdot H\left(M1\right)}$$

Eventually, after using the generated decryption key, the DU converts the desired ciphertext data into plaintext data by means of a decryption algorithm.

#### 4.4.5. Auditor Validation

In a blockchain network, the auditor relies on verification from two main aspects:

(1)Data integrity check;A blockchain uses a hash function to generate a unique identifier for the data. Auditors can verify that the data have not been tampered with by recalculating the hash value of the data. If the hash value matches, it indicates data integrity.Each block in a blockchain contains the hash value of the previous block, and this chain structure makes data tampering extremely difficult. An auditor can confirm the correctness of data by checking the integrity of the chain.The auditor can track and verify the process of data manipulation by verifying the operation log.

(2)Datetime checking.On blockchains that support smart contracts (e.g., Ether), the auditor can check the smart contract code and execution logs to ensure that the contract is executing as expected and generating the correct data.A timestamp records the exact time the data were added to the blockchain, and an auditor can verify the block’s timestamp to ensure the timeliness of the data.All nodes in the blockchain network synchronize the latest copy of the data. Auditors can check data consistency between nodes to ensure that all nodes have up-to-date data.

Using the above methods, the auditor can effectively verify the correctness and timeliness of the data in the blockchain network, thus ensuring the transparency and trustworthiness of the system.

## 5. Theoretical Analysis and Discussion

### 5.1. Correctness Analysis

In order to verify the correctness of our program, it is shown below in two steps through equations.

As the first step, to verify the correctness of the data uploaded by the data holder to the IPFS signature, the correctness of the signature is verified using the public key and the signature value downloaded from IPFS as follows.
$$\begin{array}{l} \;\;\;\lambda =e\left(g_{1}^{-\tau i\cdot \beta i},g_{2}^{\frac{rD-1}{\beta i}}g_{2}^{\frac{\left(rD+H\left(M\right)\right)P_{\left(DH_{id},s\right)}\left(\beta \right)}{\prod_{a\in S\cup D_{n+1-1\backslash DHid}}\left(\beta +\nu \left(a\right)\right)}}\right)\cdot e{\left(g_{1}^{\tau },g_{2}\right)}^{H(M1)\left(1-X_{1}-\frac{1}{X_{1}}\right)}\\ \;\;\;\;\;\cdot e(g_{1}^{\frac{rD+H\left(M\right)}{{\prod_{ai\in AA\cap A_{DH}}}^{\left(\beta +\nu \left(a\right)\right)}{\prod_{a\in S\cup D_{n+1-1\backslash DHid}}}^{\nu \left(a\right)}}},g_{2}^{\tau i\prod_{a\in S\cup D_{n+1-1\backslash DHid}}\left(\beta +\nu \left(a\right)\right)})\\ \;\;\;\;=e\left(g_{1}^{-\tau i\cdot \beta i},g_{2}^{H\left(M\right)\cdot \frac{P_{\left(DH_{id},s\right)}\left(\beta \right)}{{\prod_{a\in S\cup D_{n+1-1\backslash DHid}}}^{\nu \left(a\right)}}}\right)\cdot e\left(g_{1}^{-\tau i\cdot \beta i},g_{2}^{\frac{rD-1}{\beta i}}g_{2}^{rD\cdot \frac{P_{\left(DH_{id},s\right)}\left(\beta \right)}{\prod_{a\in S\cup D_{n+1-1\backslash DHid}}\left(\beta +\nu \left(a\right)\right)}}\right)\\ \;\;\;\;\;\cdot e{\left(g_{1}^{\tau i},g_{2}\right)}^{H\left(M\right)}\cdot e{\left(g_{1}^{\tau i},g_{2}\right)}^{H\left(M\right)\cdot X_{1}}\cdot e{\left(g_{1}^{\tau i},g_{2}\right)}^{\frac{-H\left(M\right)}{X_{1}}}\\ \;\;\;\;\;\cdot e\left(g_{1}^{\frac{rD}{{\prod_{ai\in AA\cap A_{DH}}}^{\left(\beta +\nu \left(a\right)\right)}{\prod_{a\in S\cup D_{n+1-1\backslash DHid}}}^{\nu \left(a\right)}}},g_{2}^{\tau i\prod_{a\in S\cup D_{n+1-1\backslash DHid}}\left(\beta +\nu \left(a\right)\right)}\right)\\ \;\;\;\;\;\cdot e\left(g_{1}^{\frac{H\left(M\right)}{{\prod_{ai\in AA\cap A_{DH}}}^{\left(\beta +\nu \left(a\right)\right)}{\prod_{a\in S\cup D_{n+1-1\backslash DHid}}}^{\nu \left(a\right)}}},g_{2}^{\tau i\prod_{a\in S\cup D_{n+1-1\backslash DHid}}\left(\beta +\nu \left(a\right)\right)}\right)\\ \;\;\;\;=e{\left(g_{1}^{\tau i},g_{2}\right)}^{H\left(M\right)}\cdot {\left(g_{1}^{\tau i},g_{2}\right)}^{-H\left(M\right)\cdot X_{1}}\cdot e{\left(g_{1}^{\tau i},g_{2}\right)}^{\frac{-H\left(M\right)}{X_{1}}}\cdot \left(g_{1}^{\tau i},g_{2}\right)\\ \;\;\;\;\;\cdot e{{\left(g_{1}^{\tau i},g_{2}\right)}^{H\left(M\right)\cdot \prod }}_{{\prod_{a\in S\cup D_{n+1-1\backslash DHid}}}^{{\nu \left(a\right)}^{\frac{\left(\beta +\nu \left(a\right)\right)}{\nu \left(a\right)}}}}\cdot e{{\left(g_{1}^{\tau i},g_{2}\right)}^{H\left(M\right)\cdot \prod }}_{{\prod_{a\in S\cup D_{n+1-1\backslash DHid}}}^{{\nu \left(a\right)}^{\frac{\nu \left(a\right)}{\nu \left(a\right)}}}}\\ \;\;\;\;\;\cdot e(g_{2}^{\frac{H\left(M\right)\cdot {\prod_{a\in S\cup D_{n+1-1\backslash DHid}}}^{\left(\beta +\nu \left(a\right)\right)}}{{\prod_{ai\in AA\cap A_{DH}}}^{\left(\beta +\nu \left(a\right)\right)}{\prod_{a\in S\cup D_{n+1-1\backslash DHid}}}^{\nu \left(a\right)}}},g_{1}^{\tau i})\\ \;\;\;\;=e\left(g_{1}^{\tau i},g_{2}\right) \end{array}$$

Next, after authentication is passed, the data are decrypted with the decryption key with the following equation:$$\begin{array}{l} M^{\prime }=M⊕\lambda^{s}⊕\left(\frac{e\left(\eta ,g_{2}^{\left(\tau +r_{1}\right)/\beta }\right)}{e{\left(g_{1},g_{2}\right)}^{r_{1}s}}\right)\\ \;\;=M⊕e{\left(g_{1},g_{2}\right)}^{\tau s}⊕\left(\frac{e\left(\left(g_{1}^{\beta s}\right),g_{2}^{\left(\tau +r_{1}\right)/\beta }\right)}{e{\left(g_{1},g_{2}\right)}^{r_{1}s}}\right)\\ \;\;=M⊕e{\left(g_{1},g_{2}\right)}^{\tau s}⊕\left(\frac{e{\left(g_{1},g_{2}\right)}^{\beta s\cdot \left(\tau +r_{1}\right)/\beta }}{e{\left(g_{1},g_{2}\right)}^{r_{1}s}}\right)\\ \;\;=M⊕e{\left(g_{1},g_{2}\right)}^{\tau s}⊕\left(\frac{e{\left(g_{1},g_{2}\right)}^{\left(\tau s+r_{1}s\right)}}{e{\left(g_{1},g_{2}\right)}^{r_{1}s}}\right)\\ \;\;=M⊕e{\left(g_{1},g_{2}\right)}^{\tau s}⊕\left(\frac{e{\left(g_{1},g_{2}\right)}^{r_{1}s}{\left(g_{1},g_{2}\right)}^{\tau s}}{e{\left(g_{1},g_{2}\right)}^{r_{1}s}}\right)\\ \;\;=M⊕e{\left(g_{1},g_{2}\right)}^{\tau s}⊕e{\left(g_{1},g_{2}\right)}^{\tau s}\\ \;\;=M \end{array}$$

Thus, the content of the required data is obtained.

### 5.2. Security Analysis

For the purpose of this subsection, we analyzed the safety of our proposed scheme, of which the security proof construction relies on attribute-based signature encryption and ciphertext policy attribute-based encryption. Meanwhile, the security proof of our suggested scheme refers to schemes FCP-ABSC [51]. In the following, we mainly focus on four aspects of security theory: confidentiality, unforgeability, verifiability, and privacy.

#### 5.2.1. Confidentiality Analysis

**Theorem** **1.** *An adversary is considered to be secure under the IND-CCA model for scheme B if it does not have a probabilistic polynomial time for symmetric encryption and decryption or attribute-based signing*.

**Proof** **of** **Theorem** **1.** Assuming that we construct an adversary A and that A can crack our proposed scheme B with a significantly higher probability of success than a randomized attack, the detailed construction process of A is described as follows.Initialization: The attacker A chooses a particular set of attribute sets S* when performing an attack and informs the challenger C about them for interaction.Setup: The challenger C performs Algorithms Global.Setup and AuthoritySetup. First Algorithm AuthoritySetup chooses a third-order cyclic linear group $G_{1}$, $G_{2}$, and $G_{T}$, all of prime order P. In addition, the generating elements of $G_{1}$ and $G_{2}$ are $g_{1}$ and $g_{2}$, respectively. Two collusion-resistant hash functions are chosen at random: $H_{0}:{\left\{0,\left.1\right\}\right.}^{*}\to Z_{p}$, $H_{1}:{\left\{0,\left.1\right\}\right.}^{*}\to G_{1}$. And a specific encoding function $\nu :U\to Z_{p}$, where U is an attribute domain containing a set of attributes, and $\left|\left.U\right|\right.=q$, q denotes the number of attributes. Then, Pick a set $F=\left\{f_{1},\right.f_{2},\cdots ,\left.f_{j-1}\right\}$, in which j+1≤q, pairs of distinct factors in ${\left(Z/pZ\right)}^{*}$. In addition, τ,β are randomly selected from $Z_{p}^{*}$ and calculate η,λ. And, derive the public key PK, and obtain $PK=\nu \{{G_{1},G_{2},G_{T},e,\{g_{2}^{\tau \beta^{j}}\}}_{\left\{j=0,\ldots ,2q-1\right\}},\eta ,F,\nu ,\lambda ,H_{0}\}$ by computation. At last, challenger C passes PK to A.Phase1: During this phase, challenger keeps blank tabs for lists and adversary A requests the following query. First, A requests a query for a set of private keys with attribute S to be sent to challenger C to generate the private key $SK_{A}$, which is: $SK_{A}=({\left\{g_{1}^{\frac{r}{\beta +\nu \left(a\right)}}\right\}}_{a\in S},{\left\{g_{2}^{r\beta^{i}}\right\}}_{j=0,\ldots ,q-2},g_{2}^{\frac{r-1}{\beta }})$. And then, the generated private key $SK_{A}$ is passed back to A. Next, A requests the signing of the message $M^{*}$ for signcrypt, and the challenger C executes the Keygen algorithm to generate the private key $SK_{C}$. Then, C executes $Signcrypt\left(PK,SK,\;M\;\right)$ algorithm to compute. Finally, C get the signcryption ciphertext $SCt^{*}$, which is passed to A. In the decsigncryption query phase, A makes a request to C to decrypt the signed ciphertext SCt. In the first step, C validates the attribute set $A^{*}$ that is presented by A first, and decrypts the ciphertext after passing the validation. However, C can’t demystify the ciphertext if the validation fails.Challenge: The A chooses messages $M_{0}$ and $M_{1}$ of equal length ($\left|M_{\;0}\right|=\left|M_{\;1}\right|$) and a specific attribute set $A^{*}$ to challenge C. The challenger C randomly chooses a bit $b\in \left\{0,\;1\right\}$, signs the message M with the attribute set P, and subsequently executes the KeyGen algorithm to generate the corresponding private key SK, and then C executes the signing algorithm using the private key SK to generate $SCt^{*}$ as the challenge ciphertext to be sent to A.Phase 2: Adversary A′s second request query performs a query similar to the first stage, and it cannot perform a query on the unsigned secret.Guess: Suppose attacker A guesses b′=b, then A wins the game. At this point, $Adv=|\mathrm{Pr}[b^{\prime }=b]-1/2|$ denotes the advantage of attacker A in that game. □

#### 5.2.2. Unforgeability Analysis

**Theorem** **2.** *Under a specific attack model (selected message attacks), The scheme we put forward is adaptive message unforgeable, indicating that it possesses good security. And this program’s reliability correlates with the difficulty of the aMSE-CDH problem, specifically, the security of the scheme can be translated into the difficulty of solving the aMSE-CDH problem, thus providing a mathematical puzzle-based security guarantee*.

In this security game, an attacker A tries to forge a valid signature that conforms to a specific access policy through various strategies. Based on this, A first constructs a signature-secrecy attribute that describes what A wishes to attack; then, A generates user keys associated with these attributes through KeyGen query requests. Next, A performs queries across different policy combinations and sets of signed secret attributes with the goal of obtaining random values from the user (random values increase the difficulty of A′s attack) and signed secret data for the message M. Through the operations, the attacker A tries to gather enough information to guess or infer the secret values used in the KeyGen process and the signed secret algorithm. By using difficult mathematical problems and random numbers to ensure security in this mode of attack, the difficulty of forging a signature is elevated, and even though A tries various strategies and information, he still cannot successfully forge a valid signature.

**Proof** **of** **Theorem** **2.**The unforgeability is equivalent to the setting in the message confidentiality security game. 

Query Phase: During this phase, the adversary A asks the challenger C several times for a signed cipher query, C performs the SecretGen algorithm so as to get the $SK_{A}$ and passes it to A. Then A sends M to C, which executes the KeyGen algorithm and computes the $SK_{C}$, and then C runs the signed cipher algorithm to get the signature. Finally, the signed ciphertext SCt is computed using the signed ciphertext, R, and the formula and returned to C.

Forgery Phase: Within the scope of complying with encryption strategy and signature strategy the adversary A obtained effective signcryption ciphertext SCt and signature $\phi_{1.3}$. If A wants to win the unforgeable security game, they need to calculate valid signatures $\phi_{1.1}$ and $\phi_{1.2}$. In order to accomplish this task, the adversary needs to solve an aMSE-CDH problem, which means that the adversary must deal with the variant of computing Diffie-Hellman problem to prove that they have properties necessary to satisfy the access protocol T. □

#### 5.2.3. Verifiability Analysis

**Theorem** **3.** *The scheme is verifiable if an encryption scheme or protocol is built in a prime cyclic group and its security proof is based on the Diffie-Hellman hypothesis*.

**Proof** **of** **Theorem** **3.** The Diffie-Hellman assumes that it is difficult to calculate discrete logarithm in a given cyclic group of prime order. If this assumption holds, attackers will face great difficulties when trying to crack the encryption scheme based on this assumption. This evidence shows how this assumption ensures the security of the scheme through the interplay of Adversary A and Challenger C.Setup: C executes an algorithm to obtain master key MSK and public parameter KP, and then C forwards public key PK to A.Challenge: The challenge of A to the private key of attribute set $S_{A}^{*}\left(S_{A1},S_{A2},\ldots ,S_{A\;q}\right)$ means that A wants to obtain the private information or key associated with these attributes. The Challenger C executes user key generation algorithm to generate private key $SK_{A}$, and then returns it to A. Then A requests to signcrypt a certain challenge message $M^{*}$ that meets the attribute set, C performs signcryption algorithm to generate signcryption ciphertext $SCt^{*}$ in response to A, and then sends $SCt^{*}$ to A. Finally, C decrypts signcryption ciphertext $SCt^{*}$ as well as delivers $M^{*}$ to A.Output: A formulates an access policy T, and T requires compliance with C′s intended access policy. This shows that T is legal and can obtain the corresponding permissions in the challenger’s system. C runs the signcryption algorithm, and generates a signcrypted message $SCt_{M}^{*}$ using messages $M^{*}$ and T. Then, C tries to decrypt $SCt_{M}^{*}$ to recover message $M^{*}$. In addition, once a message is processed by the hash function, no one can accurately infer the original message content, or find two different messages hashed to the same value. Because of this property of hash function, even if an attacker obtains a signature, such as $\phi_{3}$, it can’t use this signature to forge a seemingly legitimate information. If $H\left(M^{*}\right)\neq H\left(M^{*\prime }\right)$, it is impossible for the attacker’s forged signature to match the user’s signature. This ensures the verifiability of a signature: that is, it can ensure the correctness and authenticity of the signature. □

#### 5.2.4. Privacy Analysis

**Theorem** **4.** *Our proposed scheme is private, and its detailed proof process is as follows*.

**Proof** **of** **Theorem** **4.** A selects a attributes field U which is handed over to C and then C performs setting steps that derives common parameter KP and master key MSK, then send them to the A.The opponent A picks the access policy T, two attribute sets $S_{A1}$ and $S_{A2}$ ($\left|S_{A1}\right.\cap \left.S\right|=\left|S_{A1}\cap \right.\left.S\right|=t$) which meet the access policy, and a message M to challenge C. Thereafter, C produces keys $SK_{A1}$ and $SK_{A2}$ associated with attribute sets $S_{A1}$ and $S_{A2}$, respectively, and the following is the calculation formula.
(24)$$SK_{A1}=({\left\{g_{1}^{\frac{r}{\beta +\nu \left(a\right)}}\right\}}_{a\in S_{A1}},{\left\{g_{2}^{r\beta^{i}}\right\}}_{\left\{j=0,\ldots ,q-2\right\}},g_{2}^{\frac{r-1}{\beta }})$$
(25)$$SK_{A2}=({\left\{g_{1}^{\frac{r}{\beta +\nu \left(a\right)}}\right\}}_{a\in S_{A2}},{\left\{g_{2}^{r\beta^{i}}\right\}}_{\left\{j=0,\ldots ,q-2\right\}},g_{2}^{\frac{r-1}{\beta }})$$Challenge Phase: In the beginning, A adversary requests challenger C to signcrypt message M in either of keys $SK_{A1}$ and $SK_{A2}$. Next, C randomly selects a bit b: $b\in \left\{0,\;1\left.\right\}\right.$, executes the key generation algorithm to generate the key $SK_{A,b}$, and then executes the signcryption algorithm to generate $SCt_{b}=\left(\phi_{1.1},\phi_{1.2},\phi_{1.3},\sigma_{1,1},\sigma_{1,2},\sigma_{1,3}\right)$. Afterwards, in order to verify the privacy of scheme, it is required that the signcryption values generated by challenger C using attribute set $S_{A1}$ or $S_{A2}$ are consistent. In this case, it can be concluded that the scheme is private in calculation. Specifically, the signature calculation process of adversary A is as follows.
(26)$$\begin{array}{l} SCt_{b}^{*}=e\left(g_{1}^{-\tau \cdot \beta },g_{2}^{\frac{rD-1}{\beta }}g_{2}^{\frac{\left(rD+H\left(M\right)\right)P_{\left(DH_{id},s\right)}\left(\beta \right)}{\prod_{a\in S\cup D_{n+1-1\backslash DHid}}\left(\beta +\nu \left(a\right)\right)}}\right)\cdot e{\left(g_{1}^{\tau },g_{2}\right)}^{H\left(M\right)\left(1-X_{1}-\frac{1}{X_{1}}\right)}\\ \;\;\;\cdot e\left(g_{1}^{\frac{rD+H\left(M\right)}{{\prod_{a\in AA\cap A_{DH}}}^{\left(\beta +\nu \left(a\right)\right)}{\prod_{a\in S\cup D_{n+1-1\backslash DHid}}}^{\nu \left(a\right)}}},g_{2}^{\tau \prod_{a\in S\cup D_{n+1-1\backslash DHid}}\left(\beta +\nu \left(a\right)\right)}\right)\\ \;\;=e\left(g_{1}^{\tau },\tau g_{2}\right) \end{array}$$Guess: As $\left|S_{A1}\right.\cap \left.S\right|=\left|S_{A1}\cap \right.\left.S\right|=t$, whether we use the private key $SK_{A1}$ or $SK_{A2}$ for signcryption, the signcryption values obtained are the same. It is particularly important that adversary A does not know which attribute set is used for signcryption, so our proposed scheme realizes computational privacy. □

### 5.3. Anti-Quantum Attack Analysis

Quantum computing represents an emerging computational paradigm grounded in the principles of quantum mechanics, leveraging quantum bits (qubits) over traditional bits for information processing [52]. The unique features of qubit superposition and entanglement empower quantum computers to achieve exponential speedups in specific computational tasks, particularly in complex problems such as integer factorization and discrete logarithm computations, far surpassing the capabilities of classical computers. Quantum attacks exploit the computational prowess of quantum computers to target existing cryptographic systems [53]. These attacks can potentially breach traditional cryptographic algorithms, with Shor’s algorithm [54] and Grover’s algorithm [55] standing out as classical quantum computing methodologies that demonstrate the formidable potential of quantum computing and lay the foundation for quantum cryptanalysis. Shor’s algorithm can solve complex mathematical problems, such as integer factorization and discrete logarithms, in polynomial time [54], threatening widely used public-key encryption algorithms like RSA and ECC [56], and theoretically compromising traditional asymmetric cryptography [57]. Meanwhile, Grover’s algorithm achieves a square root-level acceleration [58], which, while significant, primarily impacts symmetric encryption schemes. The actual threat of most other quantum algorithms on symmetric cryptanalysis remains uncertain, providing a buffer period for the symmetric cryptography field. By adjusting key lengths and ongoing research, existing symmetric encryption schemes can be fortified to resist quantum attacks.

In the face of these challenges, quantum-resistant properties can be harnessed through blockchain and IPFS data storage solutions. The decentralized, immutable, and transparent nature of blockchain ensures that data is distributed across multiple nodes, making it difficult for a single attacker to compromise the system. The hash chain structure, even when confronted with quantum-accelerated hash cracking, maintains high complexity, making data recovery extremely difficult [59]. Similarly, IPFS’s distributed file system indexes content through hash values, preventing data duplication and tampering. While Grover’s algorithm can accelerate single-hash cracking, IPFS’s multi-node storage complicates data recovery and tampering [60]. Additionally, attribute-based access control combined with blockchain and smart contract technology allows for dynamic access policy updates based on the current environment and user attributes. Leveraging blockchain’s decentralization and transparency, smart contracts can continuously update and optimize access policies in response to the development of quantum computing capabilities, thereby enhancing the system’s resilience to quantum attacks.

Many existing symmetric cryptographic designs have been proven to satisfy quantum provable security [61], but this does not imply that research into quantum-resistant symmetric cryptography should cease. On one hand, current security assessments of cryptographic algorithms are primarily based on idealized quantum computing models (fully mathematically defined quantum computing systems). In real quantum environments, cryptographic algorithms face numerous challenges and uncertainties. On the other hand, cryptography continually evolves alongside computational technology, and there may be unknown quantum attacks with greater performance potential [62].

Therefore, under the threat of quantum computing attacks, it is imperative to research and develop blockchain technologies resistant to quantum threats. This proactive approach can effectively guard against potential security threats posed by future quantum computing and ensure the long-term robustness and high security of blockchain systems. By integrating quantum computing with blockchain technologies, a more secure and reliable digital infrastructure can be constructed to withstand various future computational challenges, supporting new generations of secure, efficient, and reliable blockchain applications.

## 6. Evaluation and Analysis

### 6.1. Functional Analysis

Some of the existing schemes are displayed in Table 2 in comparison with our proposed scheme in terms of functional features, including signer privacy, multiple permissions, verifiability, auditing, and IPFS storage features, where “√” indicates that the feature exists in the scheme, and “×” means that the feature does not exist. In addition, we can clearly see from the table that both the schemes of Deng et al. [25] and Eltayieb et al. [8] have the signer privacy feature, while the scheme [25] also supports the authentication feature; the scheme of Yang et al. [49] is not in the IPFS storage, and in the cloud storage environment, which is not yet fully decentralized; and the two of Wang and Song’s [63] scheme does not support any of the above listed features. In contrast, our scheme has all the above features, and is more suitable for practical applications because of its excellent performance in terms of functional diversity, flexibility, and practicality.

### 6.2. Performance Analysis

#### 6.2.1. Experimental Environment

To evaluate the performance of the scheme, we conducted performance analysis from the perspectives of computational overhead and communication overhead, using simulation experiments. As shown in Table 3, the experiments were performed on an Intel i7-4900H processor with 64-bit Windows 10 operating system and 16 GB of RAM, implemented within the Pairing-Based Cryptography (PBC) library of VCCC 6.0. The experiments utilized Type A bilinear pairings [64], executing 10 trials and averaging the results. The observed outcomes were as follows: the execution time for bilinear pairing operations H=11.23 ms, the exponentiation time in group $G_{1}$, denoted as V1=5.62 ms, and the exponentiation time in group $G_{2}$, denoted as V2=1.56 ms. Additionally, a symbol $V_{T}$ was introduced to represent the exponentiation time in group $G_{T}$.

#### 6.2.2. Computational Overhead Analysis

Table 4 presents the computational costs of different schemes, primarily considering signcryption and unsigncryption. Within these schemes, the number of attributes in a signature, the number of encryptions, the number of signatures, and the number of decryption key attributes are denoted by ks, kd, Bs, and Be, respectively. To more accurately measure the computational costs across different schemes, it is assumed that ks=8, kd=6, Bs=5, and Be=3.

To enhance the readability and analytical efficiency of the data, Figure 4 vividly illustrates the differences in computational overheads among five schemes. Specifically, Scheme [49] achieves a user unsigncryption time of 2.53 ms. In contrast, our proposed scheme utilizes a signcryption key denoted as $V_{1}+V_{T}+H$, which exhibits lower computational overhead compared to Schemes [8,25,49,63]. Additionally, the proposed scheme incurs less overhead in decryption keys than Schemes [8,63], indicating that the outsourcing of computations effectively alleviates the computational burden on users, significantly reducing computational costs and enhancing efficiency.

#### 6.2.3. Communication Overhead Analysis

Table 5 compares the communication overheads among different schemes, focusing on costs associated with keys and ciphertexts. As observed in the table, the communication overheads in Schemes [8,25,63] are attribute-dependent, whereas those in Schemes [49] and our proposed scheme are not attribute-dependent. This implies that the size of the ciphertext in these schemes remains constant regardless of the number of attributes. When compared to Scheme [49], our proposed scheme exhibits relatively lower costs. Therefore, our scheme presents a more advantageous approach.

To provide a more intuitive visualization of the communication overhead differences among five distinct schemes, Figure 5 is designed to present the relevant data and trends. Through Figure 5, we can more clearly observe the data for each scheme and their relationships with the number of attributes.

Firstly, in Scheme [8], the key size is approximately 128 bytes, whereas the key sizes in Schemes [25,49,63], and ours are 640 bytes, 1216 bytes, 192 bytes, and 128 bytes, respectively. Secondly, in Scheme [8], the ciphertext size equals 320 bytes, while in Schemes [25,49,63], and ours, the ciphertext sizes are 384 bytes, 1664 bytes, 256 bytes, and 128 bytes, respectively. Finally, the experimental results demonstrate that the signature-encryption key and decryption key in our proposed scheme are independent of the number of attributes and remain constant.

Consequently, the scheme effectively addresses the challenges of communication and computational overhead. Furthermore, the scheme is a powerful yet cost-effective solution for data storage in blockchain and IPFS environments.

### 6.3. Challenges Faced

In this proposed solution, the Ethereum blockchain platform demonstrates exceptional potential in smart contracts and decentralized applications; however, it still faces significant challenges, particularly regarding scalability. The design of the Ethereum 1.0 network limits its transaction processing speed and throughput, primarily due to its Proof of Work (PoW) consensus mechanism, which results in longer transaction confirmation times and increased fees during peak periods. Although Ethereum 2.0 plans to introduce Proof of Stake (PoS) and sharding to address these issues [3], the implementation and optimization of these improvements are still ongoing, and their effectiveness requires time to assess.

Additionally, Ethereum upgrades typically necessitate broad consensus, and the coordination process can be complex. Divergences among stakeholders regarding the content and timing of upgrades may lead to protocol disputes. Furthermore, incompatible changes to protocol rules may result in network splits, also known as hard forks [65]. When hard forks occur, inconsistencies in the protocol can increase operational complexities, such as data inconsistency and divergence in transaction histories, which not only threaten the stability of the system but may also compromise the security and trustworthiness of the network.

Moreover, timely updates of expired users present another challenge. Managing expired user information and revoking privileges in a blockchain may require the design of additional mechanisms to mark and manage expired users. However, the fixed structure and immutability of the blockchain complicate the implementation of these dynamic management measures, increasing the complexity of system design. For example, the blockchain’s consensus mechanism and network latency may affect the response time for processing these changes, further impacting the timeliness of user permissions and data security.

## 7. Conclusions and Further Work

This paper presents a data sharing model based on blockchain and attribute-based signature algorithms to support multi-permission verification. The proposed solution integrates IPFS, which enhances storage efficiency by utilizing distributed storage to mitigate the risks associated with centralized storage. To address concerns regarding data reliability and trust, auditors provide online data auditing services. Furthermore, the construction of the model is based on the combination of cyclic groups and large prime numbers, effectively improving the security of cryptographic schemes and resisting various potential attacks. Rigorous security analysis demonstrates that our scheme ensures confidentiality, non-repudiation, verifiability, and privacy. Performance analysis indicates that the solution significantly reduces computational and communication overhead, making it more suitable for the storage architecture of large-scale blockchain networks. However, our approach does have limitations. Future work will focus on deploying smart contracts within consortium chains and exploring the integration or improvement of various consensus algorithms to enhance blockchain efficiency, while also investigating ways to fundamentally address storage capacity issues from the blockchain perspective.

## Figures and Tables

**Figure 1 sensors-25-00160-f001:**
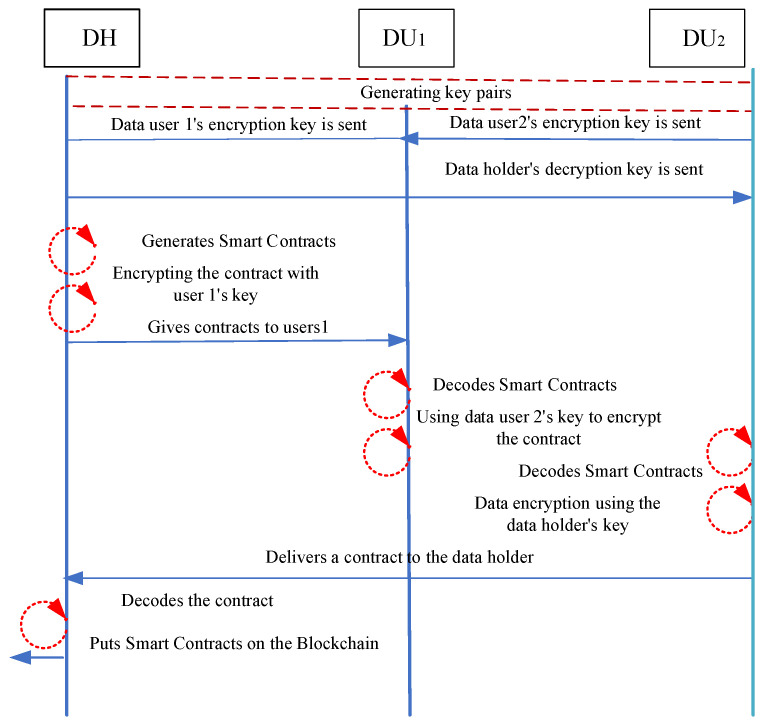
Flow of generating a smart contract.

**Figure 2 sensors-25-00160-f002:**
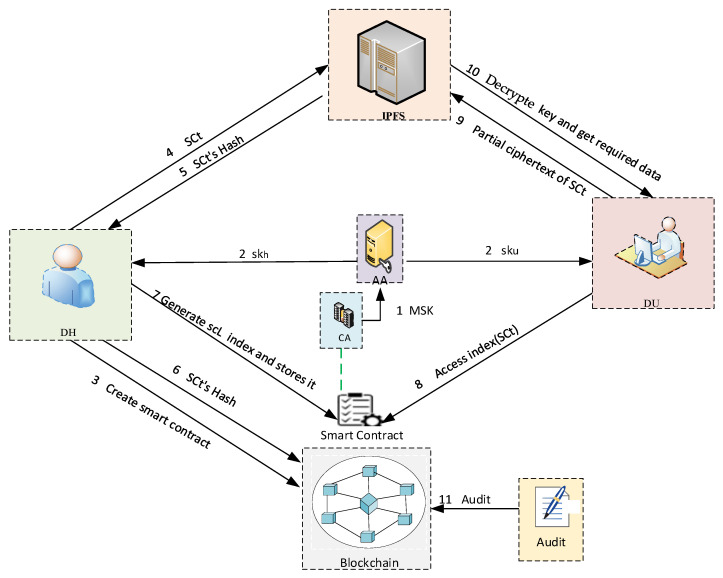
Scheme architecture.

**Figure 3 sensors-25-00160-f003:**
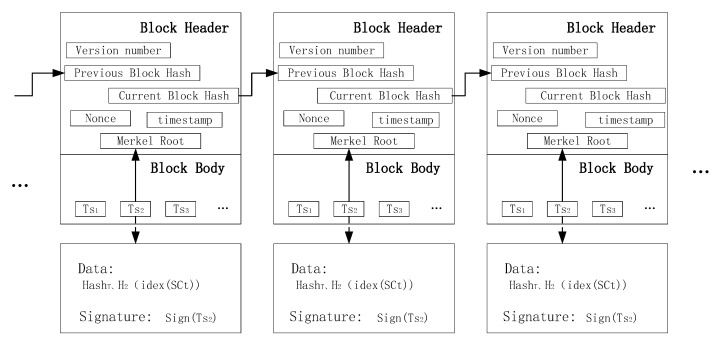
Structure of transaction information.

**Figure 4 sensors-25-00160-f004:**
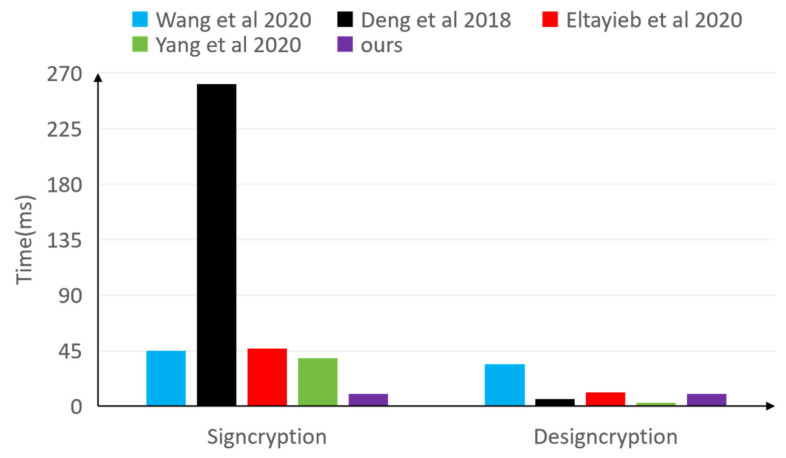
The comparison of computational expenses among the five schemes. Where: Eltayieb et al. 2020 [8]; Deng et al. 2018 [25]; Yang et al. 2020 [49]; Wang et al. 2020. [63].

**Figure 5 sensors-25-00160-f005:**
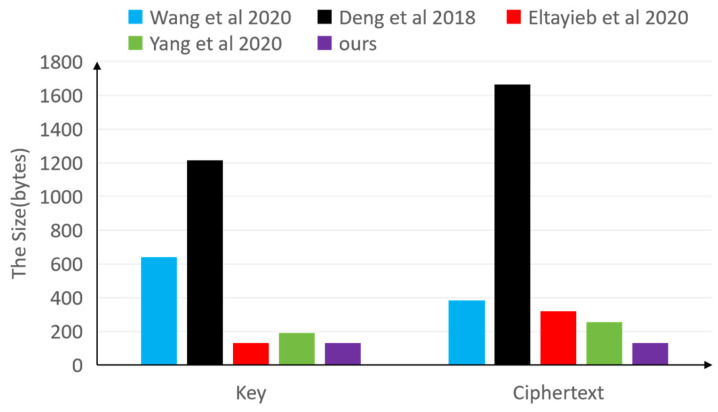
The comparison of communication expenses among the five schemes. Where: Eltayieb et al. 2020 [8]; Deng et al. 2018 [25]; Yang et al. 2020 [49]; Wang et al. 2020. [63].

**Table 1 sensors-25-00160-t001:** Symbols and description.

Symbol	Description
M	A message or data in IPFS
U	Attribute field
q	The size of the attribute field
$S_{DU}$	A set of attributes are assigned to the DU
$S_{DH}$	A set of attributes are assigned to the DH
$SK_{DH}$	The Secret Key to DH
$SK_{DU}$	The Secret Key to DU
KP	Public parameter
PK	Public key
MSK	Master secret key
AA	Attribute Authority
SCt	The signed ciphertext of message M
$Hash\left(M\right)$	The hash value of the ciphertext SCt of message M
$\mathrm{Inx}(SCt_{1})\;$	$\mathrm{Index}\;\mathrm{of}\;\mathrm{ciphertext}\;SCt_{1}$
$BlockHash_{t1}$	$\mathrm{The}\;\mathrm{block}\;\mathrm{hash}\;\mathrm{at}\;\mathrm{moment}\;t_{1}$
$E_{G1}$	$\mathrm{The}\;\mathrm{length}\;\mathrm{of}\;\mathrm{an}\;\mathrm{element}\;\mathrm{in}\;G_{1}$ in bits
$E_{G2}$	$\mathrm{The}\;\mathrm{length}\;\mathrm{of}\;\mathrm{an}\;\mathrm{element}\;\mathrm{in}\;G_{2}$ in bits
$E_{GT}$	$\mathrm{The}\;\mathrm{length}\;\mathrm{of}\;\mathrm{an}\;\mathrm{element}\;\mathrm{in}\;G_{T}$ in bits

**Table 2 sensors-25-00160-t002:** Functional comparison of some similar programs.

Scheme	Signcryptor Privacy	Multi-Agency	Verifiability	Audit	Outsourcing Calculation	IPFS
[63]	×	×	×	×	×	×
[25]	√	×	√	×	√	×
[8]	√	×	×	×	×	×
[49]	√	√	√	√	√	×
ours	√	√	√	√	√	√

**Table 3 sensors-25-00160-t003:** Experimental environment.

Experiment Setting	Allocation
Processor	Intel Core i7-4790H 3900 MHz (Santa Clara, CA, USA)
RAM	16 GB
Experiment times	10
H	11.23 ms
$V_{1}$	5.62 ms
$V_{2}$	1.56 ms

**Table 4 sensors-25-00160-t004:** Comparison of computational overheads.

Scheme	Signcryption	User Designcryption	Means
[63]	$\left(Bs+3\right)V_{1}$	3H	ABE + IBE + IBS + B
[25]	$\left(10+2Be+6Bs\right)V_{1}+V_{T}$	$V_{1}$	ABSC
[8]	$\left(2Be+2\right)V_{1}+V_{2}$	H	ABSC + B
[49]	$6V_{1}+2V_{T}$	$V_{T}$	ABSC + B
ours	$V_{1}+V_{T}+H$	$V_{1}+V_{T}+H$	ABSC + B + IPFS

**Table 5 sensors-25-00160-t005:** A comparison of communication overheads.

Scheme	Key Size	Ciphertext Size
[63]	$\left(4+kd\right)E_{G1}$	$\left(3+Be\right)E_{G1}$
[25]	$\left(ks+kd+5\right)E_{G_{1}}$	$2\left(ks+Be+2\right)E_{G_{1}}$
[8]	$2E_{G1}$	$\left(Be+2\right)E_{GT}$
[49]	$BeE_{G1}$	$\left(Be+1\right)E_{G1}$
ours	$\left(Be-1\right)E_{GT}$	$\left(Be-1\right)E_{GT}$

## Data Availability

All relevant data are included in the paper.

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
