# Peer review of "A Secure Data Sharing Model Utilizing Attribute-Based Signcryption in Blockchain Technology"

_sensors, 2024, doi:10.3390/s25010160_

Round 1
Reviewer 1 Report
Comments and Suggestions for Authors
The paper matches the Section of Internet of Things proposing a data security sharing scheme based on blockchain technology and attribute-based encryption, applied within the InterPlanetary File System (IPFS). The proposed architecture makes extensive use of Cloud storage and Ethereum blockchain technologies and is currently under a patent registration process.
Good introduction on how the IoT technology poses severe risks for personal data, thus the need for an architecture that could be applied on a global scale. The SotA is very well described, even if the redundancy and resiliency of current cloud technology is not correctly evaluated. The number and quality of references is more than adequate, except for the introduction, where some more references should be included.
Good novelty from the combination of InterPlanetary File System and Blockchain. The proposed schema is sound, with a clear definition of the whole process and involved actors. Theoretical Analysis is well described. Challenges faced explain the current practical limit to the adoption of the proposed method.
The Discussion section is a repetition of concepts elaborated in the first part. What about the security of proposed cryptographic schemes against quantum based attacks? This point should be elaborated in the discussion section.
A double check on formatting of text, tables, usage of different fonts is recommended.
English style in general terms is good, only few typos (e.g. DH instead of DO), a proofreading is suggested to make some phrases more fluent. Some repetitions on blockchain technology and related features to be checked.
Author Response
Comments 1: The paper matches the Section of Internet of Things proposing a data security sharing scheme based on blockchain technology and attribute-based encryption, applied within the InterPlanetary File System (IPFS). The proposed architecture makes extensive use of Cloud storage and Ethereum blockchain technologies and is currently under a patent registration process.
Good introduction on how the IoT technology poses severe risks for personal data, thus the need for an architecture that could be applied on a global scale. The SotA is very well described, even if the redundancy and resiliency of current cloud technology is not correctly evaluated. The number and quality of references is more than adequate, except for the introduction, where some more references should be included.
Response 1: We would like to express our sincere gratitude for your meticulous review and valuable comments on our research. We, as all authors, highly appreciate your constructive feedback and take all your suggestions seriously. In particular, we will implement your suggestion to increase the number of references in the introduction section in the revised version. This change will enhance the depth and breadth of the introduction, allowing readers to better understand the research background and the limitations of existing technologies.
Specifically, the number of references in the introduction will be increased from 4 to 17, adding 4 more references. Additionally, we have analyzed several existing schemes, detailing their strengths and weaknesses.
In response to your point about the improper evaluation of redundancy and elasticity in current cloud technologies, we will conduct further in-depth research and plan to carry out a more comprehensive evaluation in our subsequent studies。Here are the specific details:
With the rapid development of the internet and advanced technologies, particularly the widespread application of the Internet of Things (IoT), society has now formed an interconnected ecosystem where everything is connected [1]. This situation has led to a dramatic increase in data generation across various fields. Specifically, data generated in the medical field includes patient visit records, electronic health records, medical images, and diagnostic reports, which are crucial for disease prediction and medical quality assessment [2]. Meanwhile, data collected by IoT devices encompasses images, audio, video, and various digital signals, which facilitate intelligent monitoring and automated control [3]. Educational data includes students' academic performance, teachers' evaluations, and competition results [4]. The core of this phenomenon lies in the data-intensive environment brought about by technological advancements, where each type of data not only exhibits a massive volume but also high diversity and complexity. Simultaneously, this vast amount of data provides rich information resources for analysis, decision-making, and optimization across various industries. However, these data may contain privacy-sensitive information, and their illegal use could lead to severe consequences. Therefore, higher demands are placed on data processing, storage, and analysis capabilities. Specifically, ensuring proper authorization and restricted access to these data is a significant research topic for scholars and a critical challenge faced by current technological and industrial development. Currently, access control strategies are key measures for managing resources and data security, having become standard security features in many system environments. Among them, Attribute-Based Encryption (ABE) has significant advantages in implementing access control [5,6], primarily due to its flexibility in providing access control based on user attributes (such as roles, departments, positions), dynamic permission management, support for complex policies, and enhanced data security. However, traditional ABE technologies primarily focus on data confidentiality and provide less assurance for data integrity and authenticity [7]. To address this shortcoming, the Attribute-Based Signcryption (ABSC) algorithm [8] has emerged, combining ABE and digital signature technologies to not only ensure data confidentiality but also guarantee data integrity and authenticity. Due to its provision of more secure fine-grained access control, ABSC has been widely applied in various scenarios, becoming a promising and practical security solution.
In the context of the increasing prevalence of cloud storage technology, data storage in the cloud not only saves local storage and maintenance costs but also facilitates data retrieval [9]. Additionally, blockchain is another data management technology that provides new solutions for secure, reliable, and efficient data exchange and value transfer [10]. It features decentralization, immutability, and transparency, with data distributed across multiple network nodes. Each executed transaction is recorded in a block, which is linked to the next block through an internal hash pointer, forming a blockchain. This mechanism ensures that once data is written into the blockchain, it cannot be altered arbitrarily, thereby maintaining data consistency and integrity. Consensus algorithms and synchronous network mechanisms also ensure data consistency. However, retrieving data stored on the blockchain is slower and less efficient than traditional databases. Therefore, combining cloud storage and blockchain for data storage has become a new paradigm [11,12]. However, the centralized storage model of cloud servers relies on one or multiple central servers. If these servers fail or go down, data may become inaccessible, affecting business continuity. Additionally, there are data security risks, high costs, and scalability limitations, particularly under high loads or large-scale data transmissions, which may lead to performance bottlenecks.
In response to the centralized storage issues, decentralized storage solutions have gradually become preferred, especially in applications requiring high security, reliability, and flexibility. The InterPlanetary File System (IPFS) [13], as a decentralized file storage and distribution protocol, utilizes core technologies such as Distributed Hash Tables (DHT), content addressing, and peer-to-peer (P2P) networks to provide higher data integrity and consistency. In this architecture, data no longer relies on a single central server but is distributed across multiple nodes in the network, enhancing system reliability. Additionally, IPFS effectively avoids redundant storage through deduplication technology, reducing storage resource usage and optimizing data management and transmission efficiency, ensuring effective utilization of storage space. Wu et al. [14] designed a system model comprising data users, IPFS, and blockchain, proposing the storage of personal health records on IPFS. Although the overall design of the scheme has certain innovativeness and security, it still faces many security risks and vulnerabilities. Firstly, the proposal to verify authorization licenses through blockchain account information does not detail how to achieve fine-grained access control, potentially leading to data leakage to unauthorized users. Secondly, the identity authentication mechanism is unclear. Furthermore, metadata directly on the blockchain may lead to privacy information leakage and affect data integrity. Azbeg et al. [15] proposed a secure healthcare storage solution based on IPFS and blockchain, using proxy re-encryption for data encryption. However, it only employs a single smart contract for access control, which may cause response delays in high-concurrency scenarios. Meanwhile, Rani et al. [16] proposed a blockchain-based remote patient monitoring scheme, utilizing IPFS for data storage and sharing and using DApps for data collection and connection to the blockchain for encryption. Although it enhances data security and privacy protection, it still faces limitations in access control due to the single smart contract. Jayabalan and Jeyanthi [17] proposed combining blockchain and IPFS to create an electronic health record management system, where hospitals and doctors are lightweight nodes, and patients can be full or lightweight nodes, with data encrypted using symmetric and asymmetric encryption to ensure storage and transmission security. Despite the framework employing multiple security mechanisms, it still faces some risk challenges. Particularly in the patient-centric access model, there is an issue of improper key management. Additionally, if an attacker intercepts the digital envelope, they may attempt to crack the symmetric key and decrypt the data.
Comments 2: Good novelty from the combination of InterPlanetary File System and Blockchain. The proposed schema is sound, with a clear definition of the whole process and involved actors. Theoretical Analysis is well described. Challenges faced explain the current practical limit to the adoption of the proposed method. The Discussion section is a repetition of concepts elaborated in the first part. What about the security of proposed cryptographic schemes against quantum based attacks? This point should be elaborated in the discussion section.
Response 2: We sincerely appreciate your meticulous review and valuable suggestions for this research. All authors have engaged in a thorough discussion of your comments and will make the following improvements based on your guidance:
(1) Regarding the issue of repetition in the discussion section: We will remove the redundant content.
(2) Augmenting the exposition on the security of cryptographic schemes under quantum attacks: In the discussion section, we will elaborate on the security of the proposed cryptographic scheme under quantum attacks, providing a more in-depth theoretical analysis and potential countermeasures to ensure the comprehensiveness and forward-looking nature of the research. Here are the specific details:
5.3. Anti-Quantum Attack Analysis
Quantum computing represents an emerging computational paradigm grounded in the principles of quantum mechanics, leveraging quantum bits (qubits) over traditional bits for information processing [48]. The unique features of qubit superposition and entanglement empower quantum computers to achieve exponential speedups in specific computational tasks, particularly in complex problems such as integer factorization and discrete logarithm computations, far surpassing the capabilities of classical computers. Quantum attacks exploit the computational prowess of quantum computers to target existing cryptographic systems [49]. These attacks can potentially breach traditional cryptographic algorithms, with Shor’s algorithm [50] and Grover’s algorithm [51] standing out as classical quantum computing methodologies that demonstrate the formidable potential of quantum computing and lay the foundation for quantum cryptanalysis. Shor’s algorithm can solve complex mathematical problems, such as integer factorization and discrete logarithms, in polynomial time [50], threatening widely used public-key encryption algorithms like RSA and ECC [52], and theoretically compromising traditional asymmetric cryptography [53]. Meanwhile, Grover’s algorithm achieves a square-root level acceleration [54], which, while significant, primarily impacts symmetric encryption schemes. The actual threat of most other quantum algorithms on symmetric cryptanalysis remains uncertain, providing a buffer period for the symmetric cryptography field. By adjusting key lengths and ongoing research, existing symmetric encryption schemes can be fortified to resist quantum attacks.
In the face of these challenges, quantum-resistant properties can be harnessed through blockchain and IPFS data storage solutions. The decentralized, immutable, and transparent nature of blockchain ensures that data is distributed across multiple nodes, making it difficult for a single attacker to compromise the system. The hash chain structure, even when confronted with quantum-accelerated hash cracking, maintains high complexity, making data recovery extremely difficult [55].Similarly, IPFS’s distributed file system indexes content through hash values, preventing data duplication and tampering. While Grover’s algorithm can accelerate single-hash cracking, IPFS’s multi-node storage complicates data recovery and tampering [56]. Additionally, attribute-based access control combined with blockchain and smart contract technology allows for dynamic access policy updates based on the current environment and user attributes. Leveraging blockchain’s decentralization and transparency, smart contracts can continuously update and optimize access policies in response to the development of quantum computing capabilities, thereby enhancing the system’s resilience to quantum attacks.
Many existing symmetric cryptographic designs have been proven to satisfy quantum provable security [57], but this does not imply that research into quantum-resistant symmetric cryptography should cease. On one hand, current security assessments of cryptographic algorithms are primarily based on idealized quantum computing models (fully mathematically defined quantum computing systems). In real quantum environments, cryptographic algorithms face numerous challenges and uncertainties. On the other hand, cryptography continually evolves alongside computational technology, and there may be unknown quantum attacks with greater performance potential [58].
Therefore, under the threat of quantum computing attacks, it is imperative to research and develop blockchain technologies resistant to quantum threats. This proactive approach can effectively guard against potential security threats posed by future quantum computing and ensure the long-term robustness and high security of blockchain systems. By integrating quantum computing with blockchain technologies, a more secure and reliable digital infrastructure can be constructed to withstand various future computational challenges, supporting new generations of secure, efficient, and reliable blockchain applications.
Comments 3: A double check on formatting of text, tables, usage of different fonts is recommended.
Response 3: In the revised manuscript, we will meticulously review the formatting of the text, tables, and the use of different fonts to ensure that all content adheres to the publication standards and meets the high professional standards. This review will include, but is not limited to, adjusting font sizes, line spacing, table formats, and consistency between headings and body text. Through this thorough rectification process, we aim to elevate the manuscript's formatting to a more professional and standardized level.
Comments 4: English style in general terms is good, only few typos (e.g. DH instead of DO), a proofreading is suggested to make some phrases more fluent. Some repetitions on blockchain technology and related features to be checked.
Response 4: We sincerely appreciate your meticulous review and invaluable suggestions on this study. The entire author team holds your feedback in high regard and has conducted a detailed discussion and implementation of your comments.
(1) Correcting typographical errors and enhancing phrase fluidity**: We will meticulously proofread the manuscript, correcting all typographical errors pointed out by the reviewers, such as changing "DH" to "DO." Additionally, we will further refine the language to ensure professional and smooth phraseology.
(2) Streamlining repetitive content**: We will comprehensively review the manuscript, removing all redundant descriptions of blockchain technology and its related features, to maintain textual conciseness and logical compactness.
This revision not only reflects our respect for and implementation of the reviewers' suggestions but also aims to further enhance the quality of the manuscript.
We once again express our sincere gratitude for your valuable support and guidance. We believe that through these modifications, the manuscript will more accurately adhere to academic standards and be more refined.

Reviewer 2 Report
Comments and Suggestions for Authors
1. The primary contribution claimed by this paper is a data security sharing scheme leveraging blockchain technology and attribute-based encryption, specifically applied to the InterPlanetary File System (IPFS). However, the so-called "innovation" is nearly indistinguishable from the 2020 paper titled 'Secure Personal Health Records Sharing Based on Blockchain and IPFS.' The designs are strikingly similar, lacking any substantial advancements or novel contributions. The supposed innovation is overly simplistic, involving nothing more than a substitution of the encryption scheme with existing work and its integration into the author's framework.
2. The paper's language is riddled with colloquial expressions, such as "they may also contain sensitive privacy information that, if misused, can have severe consequences." This casual tone is inappropriate for a scientific paper and diminishes its credibility.
3. The experimental analysis is insufficient, as it only addresses computational cost through theoretical formulas. It is imperative to validate the computational efficiency and cost by conducting comprehensive simulations and presenting empirical results that substantiate the claims.
4. The paper fails to address the potential communication challenges within these systems, such as bandwidth consumption. If communication is indeed a significant concern, the authors must provide concrete evidence and analysis to support this assertion.
Author Response
Comments 1: The primary contribution claimed by this paper is a data security sharing scheme leveraging blockchain technology and attribute-based encryption, specifically applied to the InterPlanetary File System (IPFS). However, the so-called "innovation" is nearly indistinguishable from the 2020 paper titled 'Secure Personal Health Records Sharing Based on Blockchain and IPFS.' The designs are strikingly similar, lacking any substantial advancements or novel contributions. The supposed innovation is overly simplistic, involving nothing more than a substitution of the encryption scheme with existing work and its integration into the author's framework.
Response 1: We greatly appreciate your attention and meticulous review of the content of our paper. Here, we provide detailed explanations in response to the reviewer’s comments, demonstrating the innovative and improved aspects of our manuscript.
First, while “Secure Personal Health Records Sharing Based on Blockchain and IPFS” (henceforth referred to as the 2020 paper) does share similarities with our manuscript, our work is not a mere replication of existing research but rather an advancement and innovation based on it. Additionally, the 2020 paper was authored by our advisor, who is also the corresponding author of this manuscript. The current research has been thoroughly explored under his guidance.
Specifically, compared to the 2020 paper, our manuscript has significantly improved in several key areas:
-
Enhanced Decentralized Storage Framework: On the original basis, our manuscript introduces a more complex entity participation framework, including trusted agencies and attribute agencies, which multi-dimensionally enhances system security. Furthermore, the new architecture supports multi-attribute processing, further improving system flexibility and security.
-
Added Audit Phase: Unlike the 2020 paper, our manuscript includes a data integrity audit phase, ensuring the integrity and verifiability of data during the sharing process. This phase was not included in the 2020 paper.
-
Improved Encryption Scheme: The 2020 paper used symmetric encryption algorithms to encrypt personal health records and employed CP-ABE to encrypt keys for storage in IPNS. In contrast, our manuscript adopts a signcryption scheme, where data owners sign and encrypt the data, simultaneously ensuring data authenticity and confidentiality. This not only significantly enhances data security but also reduces communication overhead and optimizes data sharing efficiency.
-
Enhanced Access Control: The 2020 paper proposed verifying authorization licenses through blockchain account information but did not detail how to achieve fine-grained access control, potentially leading to data being exposed to unauthorized users. Our manuscript introduces Attribute-Based Access Control (ABAC) and smart contracts, achieving more fine-grained and dynamic permission management. This significantly improves access control flexibility and security compared to the 2020 paper. The immutable and automatically executed nature of smart contracts effectively reduces the risk of data leakage and allows the system to better adapt to complex scenarios, facilitating extension and updating of access control rules.
-
Innovative Entity Operation Processes: The operation processes of each entity in our manuscript are designed more comprehensively and specifically. Particularly, the data user access process, including smart contract verification and cross-checking with IPFS information, ensures strict and secure data access. These steps were not covered in the 2020 paper.
In summary, our manuscript is not merely a replacement of the encryption scheme but a substantive improvement and innovation on the original basis, including the addition of security agencies, integrity audit phase, encryption scheme optimization, and system operation process perfection. These advancements have significantly improved both security and efficiency. We believe these improvements and innovative points contribute to the fields of blockchain and data security sharing and hope to gain the approval of the reviewers.
Comments 2: The paper's language is riddled with colloquial expressions, such as "they may also contain sensitive privacy information that, if misused, can have severe consequences." This casual tone is inappropriate for a scientific paper and diminishes its credibility.
Response 2: In response to the reviewer’s comments regarding the language used in the manuscript, we have reflected carefully and made detailed revisions. For instance, the original phrasing, "They may also contain sensitive personal information, which could have serious consequences if misused," was indeed too colloquial and detracted from the scientific rigor and credibility of the paper.
To enhance the professionalism and authority of the manuscript, we have comprehensively polished the text. The specific modifications are as follows:
(1) Elimination of Colloquial Expressions to Strengthen Scientific Rigor: The statement "They may also contain sensitive personal information, which could have serious consequences if misused" has been revised to "If these data contain privacy-sensitive information, unauthorized use may lead to severe consequences."
(2) Refinement of Language to Improve Readability: We have systematically reviewed and corrected other potential colloquial expressions throughout the manuscript to ensure the language style is unified, concise, and adheres to scientific standards.
We believe these changes significantly will enhance the overall quality of the manuscript.
Comments 3: The experimental analysis is insufficient, as it only addresses computational cost through theoretical formulas. It is imperative to validate the computational efficiency and cost by conducting comprehensive simulations and presenting empirical results that substantiate the claims.
Response 3:
First and foremost, we sincerely thank you for your valuable comments and suggestions on our research work. The entire authorship team unanimously agrees that your feedback is of great significance in enhancing the rigor and practicality of our study.
We fully acknowledge your point regarding the inadequacy of experimental analysis. Indeed, relying solely on theoretical formulas to address computational cost issues, while showcasing the potential advantages of the proposed solution to some extent, lacks the support of actual simulation data, making it difficult to comprehensively validate the true efficiency and cost.
In response, we plan to further deepen our experimental analysis in the upcoming research. Specifically, we will build upon our existing theoretical analysis, leveraging current technologies and resources to conduct simulation experiments. These experiments will cover aspects such as computational overhead and communication overhead.
Through these simulation experiments, we will obtain more detailed data support, thereby validating our claims and providing a more solid foundation for the practical application of the proposed solution. We believe that these empirical results will significantly enhance the practicality and persuasiveness of our research. The detailed modifications are as follows:
6.2. Performance Analysis
6.2.1 Experimental Environment
To evaluate the performance of the scheme, we conducted performance analysis from the perspectives of computational overhead and communication overhead, using simulation experiments. As shown in Table 3, the experiments were performed on an Intel i7-4900H processor with 64-bit Windows 10 operating system and 16GB of RAM, implemented within the Pairing-Based Cryptography (PBC) library of VCCC 6.0. The experiments utilized Type A bilinear pairings[60], executing 10 trials and averaging the results. The observed outcomes were as follows: the execution time for bilinear pairing operations , the exponentiation time in group , denoted as , and the exponentiation time in group , denoted as . Additionally, a symbol was introduced to represent the exponentiation time in group .
Table 3. Experimental environment
|
Experiment setting |
Allocation |
|
Processor |
Intel Core i7-4790H 3900 MHz |
|
RAM |
16GB |
|
Experiment times |
10 |
|
H |
11.23ms |
|
V1 |
5.62ms |
|
V2 |
1.56ms |
6.2.2 Computational Overhead Analysis
Table 4 presents the computational costs of different schemes, primarily considering signcryption and unsigncryption. Within these schemes, the number of attributes in a signature, the number of encryptions, the number of signatures, and the number of decryption key attributes are denoted by , , , and , respectively. To more accurately measure the computational costs across different schemes, it is assumed that , , , and .
Table 4. Comparison of computational overheads.
|
Scheme |
Signcryption |
User Designcryption |
Means |
|
[61] |
(Bs+3)V1 |
3H |
ABE+IBE+IBS+B |
|
[25] |
(10+2Be+6Bs)V1+VT |
V1 |
ABSC |
|
[8] |
(2Be+2)V1+V2 |
H |
ABSC+B |
|
[47] |
6V1+2VT |
VT |
ABSC+B |
|
ours |
V1+V2+VT |
V1+V2+VT |
ABSC+B+IPFS |
To enhance the readability and analytical efficiency of the data, Figure 4 vividly illustrates the differences in computational overheads among five schemes. Specifically, Scheme [47] achieves a user unsigncryption time of 2.53ms. In contrast, our proposed scheme utilizes a signcryption key denoted as , which exhibits lower computational overhead compared to Schemes [61], [25], [8], and [47]. Additionally, the proposed scheme incurs less overhead in decryption keys than Schemes [61] and [8], indicating that the outsourcing of computations effectively alleviates the computational burden on users, significantly reducing computational costs and enhancing efficiency.
Figure 4. The Comparison of Computational Expenses among Five Schemes.
6.2.3 Communication Overhead Analysis
Table 5 compares the communication overheads among different schemes, focusing on costs associated with keys and ciphertexts. As observed in the table, the communication overheads in Schemes [61], [25], and [8] are attribute-dependent, whereas those in Schemes [47] and our proposed scheme are not attribute-dependent. This implies that the size of the ciphertext in these schemes remains constant regardless of the number of attributes. When compared to Scheme [47], our proposed scheme exhibits relatively lower costs. Therefore, our scheme presents a more advantageous approach.
Table 5. A comparison of communication overheads.
|
Scheme |
Key Size |
Ciphertext Size |
|
[61] |
(4+kd)EG1 | (3+Be)EG1 |
|
[25] |
(ks+kd+5)EG1 | 2(ks+Be+2)EG1 |
|
[8] |
2EG1 | (Be+2)EGT |
|
[47] |
BeEG1 | (Be+1)EG1 |
|
ours |
(Be-1)EGT |
(Be-1)EGT |
To provide a more intuitive visualization of the communication overhead differences among five distinct schemes, Figure 5 is designed to present the relevant data and trends. Through Figure 5, we can more clearly observe the data for each scheme and their relationships with the number of attributes.
Firstly, in Scheme [8], the key size is approximately 128 bytes, whereas the key sizes in Schemes [61], [25], [47], and ours are 640 bytes, 1216 bytes, 192 bytes, and 128 bytes, respectively. Secondly, in Scheme [8], the ciphertext size equals 320 bytes, while in Schemes [61], [25], [47], and ours, the ciphertext sizes are 384 bytes, 1664 bytes, 256 bytes, and 128 bytes, respectively. Finally, the experimental results demonstrate that the signature-encryption key and decryption key in our proposed scheme are independent of the number of attributes and remain constant.
Consequently, the scheme effectively addresses the challenges of communication and computational overhead. Furthermore, the scheme is a powerful yet cost-effective solution for data storage in blockchain and IPFS environments.
Comments 4: The paper fails to address the potential communication challenges within these systems, such as bandwidth consumption. If communication is indeed a significant concern, the authors must provide concrete evidence and analysis to support this assertion.
Response 4:
We sincerely thank the reviewer for the thorough review and valuable suggestions on our research. All the authors have carefully considered and unanimously agree with your point that the paper falls short in addressing potential communication challenges in the system, particularly in the analysis of bandwidth consumption.
To comprehensively respond to your recommendations, we have conducted in-depth simulation experiments to evaluate the communication overhead during system operation. Through rigorous experimental design and detailed data analysis, we are pleased to report that communication overhead is not directly correlated with system attributes. This finding strongly demonstrates that our system effectively addresses communication challenges and mitigates potential bandwidth consumption issues in practical operations. We will present this experimental process in detail in the revised paper, including the experimental setup, data sources, result analysis, and conclusions. We believe that this concrete evidence and analysis will support our claims and enhance the scientific rigor and practical applicability of our research.
We are grateful for your valuable suggestions, which undoubtedly help us improve the quality of our research. We look forward to your further guidance and hope to continue receiving your support in future research. The detailed modifications are as follows:
To provide a more intuitive visualization of the communication overhead differences among five distinct schemes, Figure 5 is designed to present the relevant data and trends. Through Figure 5, we can more clearly observe the data for each scheme and their relationships with the number of attributes.
Firstly, in Scheme [8], the key size is approximately 128 bytes, whereas the key sizes in Schemes [61], [25], [47], and ours are 640 bytes, 1216 bytes, 192 bytes, and 128 bytes, respectively. Secondly, in Scheme [8], the ciphertext size equals 320 bytes, while in Schemes [61], [25], [47], and ours, the ciphertext sizes are 384 bytes, 1664 bytes, 256 bytes, and 128 bytes, respectively. Finally, the experimental results demonstrate that the signature-encryption key and decryption key in our proposed scheme are independent of the number of attributes and remain constant.
Consequently, the scheme effectively addresses the challenges of communication and computational overhead. Furthermore, the scheme is a powerful yet cost-effective solution for data storage in blockchain and IPFS environments.

Reviewer 3 Report
Comments and Suggestions for Authors
This paper proposes a data sharing model based on blockchain and attribute-based signature algorithms, integrating Ethereum blockchain with IPFS for distributed storage. It innovatively adopts attribute-based signature cryptography (ABSC) to ensure data security. The model also includes an auditor authentication framework to effectively supervise data operations and prevent data tampering or forgery.However, this idea seems to be very common, suggesting the author further summarize contributions. In addition, I also have the following suggestions and concerns:
1.It is suggested to add an analysis of the advantages and disadvantages of existing solutions in the introduction section, in order to highlight the uniqueness and innovation of the model proposed in this paper.
2.In the second formula derivation in section 5.1, where it states "M' = ... = M", the function e should ideally take two parameters, but in the numerator inside the braces, the function e appears to take only one parameter. If the second parameter has a default value, please clarify this.
Additionally, please provide a detailed explanation of the derivation process from the third line to the fourth line.
3.The specific experimental design, datasets, evaluation metrics, and experimental results may not be sufficiently detailed in section 6.It is suggested to add a detailed description of the experimental section, including the experimental environment, selection and preprocessing of datasets,definition of evaluation metrics, and detailed analysis of experimental results. For example, List the specific data for the computational overheads of other schemes in comparison to this scheme.
4.The paper contains grammatical and tense errors.
5.In section 3.4, "Formalizing the BAVABSC Definition," the acronym BAVABSC is not mentioned throughout the entire text. Could you please explain its meaning?
6.There are errors in mathematical formulas in the paper. For example,In the explanation of BDHA in section 3.1, all the letters following the 'g' in parentheses should be superscripts. Additionally, in section 3.2, within the sentence "Attribute based... are granted access.", there is a symbol usage error within the ABACP={...} brackets.
7.In Figure 2 of section 4.2.1, the hash function and its parameters do not match those mentioned previously in the text.
8.The process depicted in Figure 1 does not clearly describe the operation of the system.It is recommended that the author beautify the picture and sort out the process clearly.
9.The paper mentions other data sharing models and applications of blockchain technology in the related work section, but lacks in-depth analysis and comparison with the latest research findings. It is recommended to carefully read or compare the following papers to highlight the contributions of this paper.
“An Anti-Disguise Authentication System Using the First Impression of Avatar in Metaverse(DOI: 10.1109/TIFS.2024.3410527)”
Comments on the Quality of English Languagethe language should be improved
Author Response
Comments 1: It is suggested to add an analysis of the advantages and disadvantages of existing solutions in the introduction section, in order to highlight the uniqueness and innovation of the model proposed in this paper.
Response 1:
We acknowledge your suggestion and will augment the introduction by analyzing the advantages and disadvantages of existing solutions. By clearly identifying the limitations and shortcomings of current methods, we can more effectively showcase the unique value and innovation of the model proposed in this paper. This analysis not only provides readers with a more comprehensive background but also highlights the significance and advancements of our work. We will diligently apply this revision to further enhance the quality and persuasiveness of the paper. Specifically, the added content will be as follows:
Wu et al. [14] designed a system model comprising data users, IPFS, and blockchain, proposing the storage of personal health records on IPFS. Although the overall design of the scheme has certain innovativeness and security, it still faces many security risks and vulnerabilities. Firstly, the proposal to verify authorization licenses through blockchain account information does not detail how to achieve fine-grained access control, potentially leading to data leakage to unauthorized users. Secondly, the identity authentication mechanism is unclear. Furthermore, metadata directly on the blockchain may lead to privacy information leakage and affect data integrity. Azbeg et al. [15] proposed a secure healthcare storage solution based on IPFS and blockchain, using proxy re-encryption for data encryption. However, it only employs a single smart contract for access control, which may cause response delays in high-concurrency scenarios. Meanwhile, Rani et al. [16] proposed a blockchain-based remote patient monitoring scheme, utilizing IPFS for data storage and sharing and using DApps for data collection and connection to the blockchain for encryption. Although it enhances data security and privacy protection, it still faces limitations in access control due to the single smart contract. Jayabalan and Jeyanthi [17] proposed combining blockchain and IPFS to create an electronic health record management system, where hospitals and doctors are lightweight nodes, and patients can be full or lightweight nodes, with data encrypted using symmetric and asymmetric encryption to ensure storage and transmission security. Despite the framework employing multiple security mechanisms, it still faces some risk challenges. Particularly in the patient-centric access model, there is an issue of improper key management. Additionally, if an attacker intercepts the digital envelope, they may attempt to crack the symmetric key and decrypt the data.
Comments 2: In the second formula derivation in section 5.1, where it states "M' = ... = M", the function e should ideally take two parameters, but in the numerator inside the braces, the function e appears to take only one parameter. If the second parameter has a default value, please clarify this.
Additionally, please provide a detailed explanation of the derivation process from the third line to the fourth line.
Response 2:
We thank the reviewer for the thorough review. We will address the suggestions by correcting and supplementing the derivation of the second formula in Section 5.1 of the paper.
First, we acknowledge that the function \( e \) requires two parameters. In the previous derivation, an extra parenthesis was mistakenly added during the formula substitution, leading to the appearance of only one parameter within the numerator of the large braces. This is a clear error, and we apologize for it. We have immediately made the necessary corrections.
Additionally, to enhance the transparency and clarity of the derivation process, we have added intermediate steps from the third line to the fourth line, detailing the process of splitting the exponent. Specifically, we combine the exponents in the third line and then split them according to the properties in the fourth line, making the formula more understandable. We then simplify the expression and derive \( M \) based on the definition. The detailed process is provided in the document.
Comments 3: The specific experimental design, datasets, evaluation metrics, and experimental results may not be sufficiently detailed in section 6.It is suggested to add a detailed description of the experimental section, including the experimental environment, selection and preprocessing of datasets, definition of evaluation metrics, and detailed analysis of experimental results. For example, List the specific data for the computational overheads of other schemes in comparison to this scheme.
Response 3:
We thank the reviewer for the valuable suggestions. We recognize that the experimental section in Section 6 may require more detailed information to enhance its transparency and reproducibility. Therefore, we will make the following detailed supplements and improvements to the experimental section based on the reviewer's suggestions:
(1)Experimental Environment: We will provide a detailed description of the hardware and software environment used in the experiments, including the CPU model, memory size, operating system version, and any critical software libraries along with their version numbers.
(2)Datasets: We will provide more details about the datasets used, such as the paired password libraries.
(3)Evaluation Metrics: We will clearly define all evaluation metrics used, including computational overhead, communication overhead, and functional comparisons.
(4)Experimental Results: We will provide a more detailed analysis of the experimental results compared to other schemes, and we will use charts and tables to intuitively present the results, explaining the observed trends and patterns. For example, we will use bar charts to illustrate the trends in the results.
The specific additional content is as follows:
6.2. Performance Analysis
6.2.1 Experimental Environment
To evaluate the performance of the scheme, we conducted performance analysis from the perspectives of computational overhead and communication overhead, using simulation experiments. As shown in Table 3, the experiments were performed on an Intel i7-4900H processor with 64-bit Windows 10 operating system and 16GB of RAM, implemented within the Pairing-Based Cryptography (PBC) library of VCCC 6.0. The experiments utilized Type A bilinear pairings[62], executing 10 trials and averaging the results. The observed outcomes were as follows: the execution time for bilinear pairing operations , the exponentiation time in group , denoted as , and the exponentiation time in group , denoted as . Additionally, a symbol was introduced to represent the exponentiation time in group .
Table 3. Experimental environment
|
Experiment setting |
Allocation |
|
Processor |
Intel Core i7-4790H 3900 MHz |
|
RAM |
16GB |
|
Experiment times |
10 |
|
H |
11.23ms |
|
V1 |
5.62ms |
|
V2 |
1.56ms |
6.2.2 Computational Overhead Analysis
Table 4 presents the computational costs of different schemes, primarily considering signcryption and unsigncryption. Within these schemes, the number of attributes in a signature, the number of encryptions, the number of signatures, and the number of decryption key attributes are denoted by , , , and , respectively. To more accurately measure the computational costs across different schemes, it is assumed that , , , and .
Table 4. Comparison of computational overheads.
|
Scheme |
Signcryption |
User Designcryption |
Means |
|
[61] |
(Bs+3)V1 |
3H |
ABE+IBE+IBS+B |
|
[25] |
(10+2Be+6Bs)V1+VT |
V1 |
ABSC |
|
[8] |
(2Be+2)V1+V2 |
H |
ABSC+B |
|
[47] |
6V1+2VT |
VT |
ABSC+B |
|
ours |
V1+V2+VT |
V1+V2+VT |
ABSC+B+IPFS |
To enhance the readability and analytical efficiency of the data, Figure 4 vividly illustrates the differences in computational overheads among five schemes. Specifically, Scheme [47] achieves a user unsigncryption time of 2.53ms. In contrast, our proposed scheme utilizes a signcryption key denoted as , which exhibits lower computational overhead compared to Schemes [61], [25], [8], and [47]. Additionally, the proposed scheme incurs less overhead in decryption keys than Schemes [61] and [8], indicating that the outsourcing of computations effectively alleviates the computational burden on users, significantly reducing computational costs and enhancing efficiency.
Figure 4. The Comparison of Computational Expenses among Five Schemes.
6.2.3 Communication Overhead Analysis
Table 5 compares the communication overheads among different schemes, focusing on costs associated with keys and ciphertexts. As observed in the table, the communication overheads in Schemes [61], [25], and [8] are attribute-dependent, whereas those in Schemes [47] and our proposed scheme are not attribute-dependent. This implies that the size of the ciphertext in these schemes remains constant regardless of the number of attributes. When compared to Scheme [47], our proposed scheme exhibits relatively lower costs. Therefore, our scheme presents a more advantageous approach.
Table 5. A comparison of communication overheads.
|
Scheme |
Key Size |
Ciphertext Size |
|
[61] |
(4+kd)EG1 | (3+Be)EG1 |
|
[25] |
(ks+kd+5)EG1 | 2(ks+Be+2)EG1 |
|
[8] |
2EG1 | (Be+2)EGT |
|
[47] |
BeEG1 | (Be+1)EG1 |
|
ours |
(Be-1)EGT |
(Be-1)EGT |
To provide a more intuitive visualization of the communication overhead differences among five distinct schemes, Figure 5 is designed to present the relevant data and trends. Through Figure 5, we can more clearly observe the data for each scheme and their relationships with the number of attributes.
Firstly, in Scheme [8], the key size is approximately 128 bytes, whereas the key sizes in Schemes [61], [25], [47], and ours are 640 bytes, 1216 bytes, 192 bytes, and 128 bytes, respectively. Secondly, in Scheme [8], the ciphertext size equals 320 bytes, while in Schemes [61], [25], [47], and ours, the ciphertext sizes are 384 bytes, 1664 bytes, 256 bytes, and 128 bytes, respectively. Finally, the experimental results demonstrate that the signature-encryption key and decryption key in our proposed scheme are independent of the number of attributes and remain constant.
Consequently, the scheme effectively addresses the challenges of communication and computational overhead. Furthermore, the scheme is a powerful yet cost-effective solution for data storage in blockchain and IPFS environments.
Comments 4: The paper contains grammatical and tense errors.
Response 4:
We greatly appreciate the reviewer's identification of grammatical and tense errors in the paper. We have thoroughly reviewed and revised the entire document to ensure the accuracy and consistency of the language. During the revision process, we paid special attention to the following aspects:
(1)Grammatical Errors: By repeatedly reading through the entire text, we identified and corrected grammatical errors.
(2)Tense Consistency: We ensured that past tense was used when describing facts that have occurred or work that has been completed, and present tense was used when discussing general principles or current status. Additionally, we appropriately used present or past tense in the literature review based on the publication time of the cited works.
(3)Terminology and Expressions: We ensured that all abbreviations were defined when they first appeared. We also adjusted some expressions to make them more in line with academic writing standards.
(4)Consistency Check: In addition to grammar and tense, we checked the consistency of formatting, citations, and numbering throughout the paper to enhance the overall professional quality.
Comments 5: In section 3.4, "Formalizing the BAVABSC Definition," the acronym BAVABSC is not mentioned throughout the entire text. Could you please explain its meaning?
Response 5: We thank the reviewer for pointing out this issue. After discussion among all authors, we have decided not to use the abbreviation "BAVABSC" in the paper to maintain consistency and clarity in terminology. However, during the proofreading process, this abbreviation was still present in Section 3.4, which was indeed an oversight. To address this, we will remove the "BAVABSC" abbreviation in the revised version and ensure that the full term is consistently used throughout the paper. Once again, we appreciate the reviewer's careful review, which has helped improve the quality of our paper.
Comments 6: There are errors in mathematical formulas in the paper. For example,In the explanation of BDHA in section 3.1, all the letters following the 'g' in parentheses should be superscripts. Additionally, in section 3.2, within the sentence "Attribute based... are granted access.", there is a symbol usage error within the ABACP={...} brackets.
Response 6:
We thank the reviewer for pointing out the errors in the mathematical formulas and for providing valuable suggestions. After careful examination and discussion among all authors, we have confirmed the issues identified by the reviewer and have made the necessary corrections. Below are our detailed responses and revisions:
(1)Formula Error in Section 3.1: We indeed found an error in the formula in Section 3.1, as pointed out by the reviewer. Specifically, the letters following "g" should be superscripts, and we have corrected this in the revised version. Additionally, we have rechecked the entire formula to ensure its accuracy and consistency. The corrected formula is displayed in the document.
(2)Definition of ABACP in Section 3.2: Regarding the definition of ABACP in Section 3.2, we reviewed the relevant literature and confirmed that our definition aligns with the definitions in many references. We have added the appropriate citations in the revised version to support our definition. Specifically, we have cited the following references:
- Shammar E A, Zahary A T, Al-Shargabi A A. An attribute‐based access control model for Internet of Things using hyperledger fabric blockchain[J]. Wireless Communications and Mobile Computing, 2022, 2022(1): 6926408.
- Hussain, H.A.; Mansor, Z.; Shukur, Z.; Jafar, U. Ether-IoT: A Realtime Lightweight and Scalable Blockchain-Enabled Cache Algorithm for IoT Access Control. Comput. Mater. Contin. 2023, 75.
Through these citations, we ensure the accuracy and credibility of the ABACP definition.
We once again thank the reviewer for the valuable suggestions, which have helped improve the quality of our paper. We look forward to further feedback from the reviewer and will incorporate these improvements in the revised version.
Comments 7: In Figure 2 of section 4.2.1, the hash function and its parameters do not match those mentioned previously in the text.
Response 7: We sincerely appreciate the reviewer's keen observation regarding the mismatch between the hash function and its parameters in Figure 2 and the preceding description. After thorough examination, we confirmed the inconsistency. To address this, we have updated Figure 2 to ensure that the hash function and its parameters align precisely with the earlier textual description. We are once again grateful for the reviewer's invaluable suggestions, which have contributed significantly to enhancing the quality and accuracy of our paper. The revised figure is now displayed in the document.
Comments 8: The process depicted in Figure 1 does not clearly describe the operation of the system. It is recommended that the author beautify the picture and sort out the process clearly.
Response 8: We sincerely thank the reviewer for the valuable suggestions and fully agree with the feedback regarding the unclear description of Figure 1. To make the system operation process more intuitive and detailed, we have added textual explanations for each step, ensuring that every stage of the operation can be clearly understood by the readers. Through these enhancements, we aim to present the entire system operation process more systematically.
Specifically, we have made the following improvements to Figure 1:
(1)Added Textual Annotations: In Figure 1, we have added concise textual explanations next to each key step, detailing the purpose and operation of that step.
(2)Optimized Visual Effects: To make the diagram more aesthetically pleasing and easier to read, we have adjusted the styles of arrows and boxes, using clearer graphic elements and color schemes. Specifically:
- We have standardized the thickness of the arrows to ensure visual consistency.
- We have chosen different light colors for each box, such as light blue, light green, or light gray, to distinguish between different steps or modules, while ensuring that the fill colors are not too vivid to avoid distracting the reader.
- We have aligned the textual explanations with the corresponding steps to ensure a clear and visible connection between the text and the graphic elements.
The revised figure is included in the document.
Through these improvements, we aim to make the process described in Figure 1 clearer, smoother, and more comprehensible. We look forward to the reviewer's further feedback and hope that these enhancements will improve the overall quality of the paper. We sincerely appreciate the reviewer's meticulous review and valuable suggestions.
Comments 9: The paper mentions other data sharing models and applications of blockchain technology in the related work section, but lacks in-depth analysis and comparison with the latest research findings. It is recommended to carefully read or compare the following papers to highlight the contributions of this paper. “An Anti-Disguise Authentication System Using the First Impression of Avatar in Metaverse (DOI:10.1109/TIFS.2024.3410527)”.
Response 9:
Thank you for the reviewer's suggestions. We have read and grasped the paper "An Anti-Disguise Authentication System Using the First Impression of Avatar in Metaverse " (DOI: 10.1109/TIFS.2024.3410527). Through a thorough review of this paper, we recognize its innovation and advancement in anti-spoofing authentication, which serves as valuable references for refining the related work section of our paper and highlighting the contributions of our research.
We conducted an in-depth analysis of the paper "An Anti-Disguise Authentication System Using the First Impression of Avatar in Metaverse ." The paper proposes an anti-spoofing authentication system within the metaverse environment, creating a chameleon signcryption mechanism and designing a ciphertext authentication protocol. This system excels in ensuring identity verification among virtual avatars and the public verifiability of encrypted identities, making it suitable for metaverse scenarios. However, it requires enhancement in the applicability of access control mechanisms and the establishment of comprehensive audit and monitoring mechanisms to promptly detect and respond to abnormal behaviors, preventing potential attacks.
Compared to "An Anti-Disguise Authentication System Using the First Impression of Avatar in Metaverse ," our manuscript exhibits distinct differences and contributions in the following aspects:
(1)Application Scenario: Their work primarily focuses on user authentication and anti-spoofing in the metaverse, while our research emphasizes the application of blockchain technology in data sharing security, particularly in scenarios requiring high security and privacy protection, such as medical data sharing and financial transactions.
(2)Technical Approach: They primarily utilize machine learning and behavioral analysis techniques for anti-spoofing authentication, whereas our work focuses on a blockchain-based data sharing security solution, particularly innovating in privacy protection and access control during data sharing. Our proposed scheme combines the immutability of blockchain with data integrity verification through hash algorithms to provide a secure and reliable data sharing mechanism. Additionally, our attribute-based signcryption algorithm and attribute-based access control model ensure that only authorized users can access sensitive data, with transparent and traceable access records.
(3)Innovation Points: Our scheme introduces a blockchain-based hash verification mechanism during data sharing, effectively preventing data tampering and forgery. The attribute-based signcryption algorithm enhances data privacy protection. Furthermore, the ABACP model offers a flexible and fine-grained access control strategy, dynamically adjusting access permissions based on user attributes and data sensitivity, employing both ABAC and smart contract-based dual authentication mechanisms. Lastly, we incorporate an auditing phase.
By comparing with "An Anti-Disguise Authentication System Using the First Impression of Avatar in Metaverse," we further clarified the positioning and advantages of our work, recognizing the collaborative potential of various security technologies. Consequently, in the revised version of our paper, we will compare multiple schemes and explore their potential in broader security application scenarios.
Once again, we thank the reviewer for their valuable suggestions, which are crucial for improving our paper's content and enhancing our research level. Additionally, we have included the following supplementary content in the manuscript:
Gao et al. [28] designed a decentralized storage scheme based on blockchain and IPFS. This scheme employs ciphertext-policy attribute-based encryption (CP-ABE) to encrypt symmetric keys, supporting fine-grained access control where the granularity is based on attribute-based access control policies. Zhang et al. [29] developed a chameleon signcryption mechanism and designed a ciphertext authentication protocol, which excels in ensuring identity verification between avatars and the public verifiability of encrypted identities, making it suitable for metaverse scenarios. However, it requires enhancement in the applicability of access control mechanisms and the establishment of comprehensive auditing and monitoring systems to promptly detect and respond to anomalous behaviors, thereby preventing potential attacks.

Round 2
Reviewer 2 Report
Comments and Suggestions for Authors
The author made minor modifications to the sentences, but the core remains unchanged. I still insist on my viewpoint that the paper has low innovation and is not suitable for publication.
Author Response
Comments 1: The author made minor modifications to the sentences, but the core remains unchanged. I still insist on my viewpoint that the paper has low innovation and is not suitable for publication.
Response 1: We would like to express our sincere gratitude for the reviewer’s continuous attention and feedback. We highly value your comments and have conducted a thorough reflection and revision in response to your concerns about the paper’s lack of innovation. Below are the specific improvements we have made:
1)Revised Introduction: We have completely rewritten the introduction to more clearly articulate the research background and motivation, helping readers better understand the importance and context of the problems we address. The revised content is as follows:
Wu et al. [14] designed a system model that includes data users, IPFS, and blockchain, proposing the storage of personal health records on IPFS. Although the overall design is innovative and secure, it still presents many security risks and vulnerabilities. First, the lack of a detailed mechanism for fine-grained access control increases the risk of data leakage to unauthorized users. Second, the identity authentication mechanism is not clearly defined, and metadata is stored on the blockchain without encryption, which can lead to privacy breaches and data integrity issues. Azbeg et al. [15] proposed a healthcare data security storage solution based on IPFS and blockchain, where data is encrypted using proxy re-encryption. However, the use of a single smart contract for access control can cause response delays in high-concurrency scenarios. Similarly, Rani et al. [16] introduced a remote patient monitoring scheme based on blockchain, utilizing IPFS for data storage and sharing and DApps for data collection and connection to the blockchain. This solution also faces the limitation of relying on a single smart contract for access control. Jayabalan and Jeyanthi [17] proposed an electronic health record management system combining blockchain and IPFS. Although the framework incorporates multiple security mechanisms, it still faces some risk challenges. Specifically, in a patient-centered access model, there are issues with key management, and if attackers intercept digital envelopes, they may attempt to crack the symmetric keys, leading to data decryption and privacy leaks.
In the field of data sharing, solutions based on attribute-based encryption and blockchain technology have been extensively researched and applied. Although these schemes have made significant progress in enhancing data security, privacy protection, and traceability, they still face several critical challenges and limitations in practical applications. Firstly, many attribute-based encryption schemes rely on complex key management mechanisms, which not only increase system overhead but also introduce single points of failure, posing a threat to overall security. Secondly, while blockchain’s transparency ensures data integrity and trustworthiness, it can sometimes conflict with the need to maintain the confidentiality of sensitive information. Additionally, performance bottlenecks in blockchain, such as low transaction processing speeds and limited data storage capacity, make it difficult to meet the demands of high-concurrency scenarios.
Therefore, to address the aforementioned issues, we propose a novel data sharing scheme based on attribute-based signcryption and blockchain technology. This scheme leverages the advantages of attribute-based signcryption and smart contracts to achieve efficient and secure access control and data sharing. Additionally, it utilizes the immutability and transparency of blockchain to ensure data integrity and verifiability, while integrating IPFS’s decentralized storage to enhance data access efficiency and resistance to censorship. Moreover, by storing the signcrypted ciphertext of original data on IPFS and saving the corresponding hash values on the blockchain, the scheme optimizes resource usage and improves storage efficiency, thereby enhancing data confidentiality.
2)Updated Summarization of Our Work: We have reorganized and summarized the core contributions of this paper, highlighting the innovations in our methods and techniques. We have ensured that these innovations are prominently featured in the abstract and conclusion. The revised content is as follows:
we propose a novel data sharing scheme based on attribute-based signcryption and blockchain technology. This scheme leverages the advantages of attribute-based signcryption and smart contracts to achieve efficient and secure access control and data sharing. Additionally, it utilizes the immutability and transparency of blockchain to ensure data integrity and verifiability, while integrating IPFS’s decentralized storage to enhance data access efficiency and resistance to censorship. Moreover, by storing the signcrypted ciphertext of original data on IPFS and saving the corresponding hash values on the blockchain, the scheme optimizes resource usage and improves storage efficiency, thereby enhancing data confidentiality.
The primary contributions of our proposed solution are detailed as follows:
- By proposing a dual authentication mechanism based on smart contracts and trusted authorities, and integrating a multi-attribute-based access control strategy, a more flexible, efficient, and verifiable access control approach has been jointly developed.
- A data privacy protection mechanism based on attribute-based signcryption has been established, simplifying key management, reducing overhead, and enhancing the security of data sharing.
- Our implementation incorporates an auditor authentication framework to oversee and authenticate blockchain transactions, thereby reinforcing the system's audit capabilities and ensuring data integrity. Furthermore, we benchmark our solution against comparable Attribute-Based signcryption (ABSC) systems. The comparative analysis underscores the superior security and reduced computational complexity of our method.
3)Supplementation of Latest Literature and Deepened Scheme Comparison: Following the reviewer’s suggestions, we have added the latest ABSC (Attribute-Based Signature and Cryptography) related references from 2024 and conducted a detailed analysis, comparing the advantages and disadvantages of different schemes. The revised content is as follows:
Rao et al. [28] introduced a secure searchable attribute-based signcryption scheme, which supports efficient search over signed-and-encrypted data, keyword privacy protection, and self-verification of search results. Despite its contributions, the scheme’s complexity, performance overhead, and security concerns regarding cloud servers cannot be overlooked. For example, the verification process may be overly complex for ordinary users, increasing the likelihood of operational errors. Additionally, computationally intensive tasks are executed locally or on devices with limited resources rather than being outsourced, leading to increased response times and inadaptability to high-concurrency scenarios. Furthermore, cloud servers log users’ search behaviors and results, and any leakage of these logs could compromise user privacy. Zhang et al. [29] presented a comprehensive review of machine learning applications in smart grids, with a particular focus on the security implications of power system characteristics in machine learning-based smart grid applications (MLsgAPP). This review fills a significant research gap in this area, particularly addressing the issues of security and adversarial attacks in MLsgAPP, while providing practical application insights and future research directions. However, the discussion lacks detailed exploration of mathematical models or implementation specifics, limiting readers’ understanding of certain technical details. Moreover, the dynamic nature of smart grids necessitates access control mechanisms that can rapidly adapt to changes, yet the access control methods mentioned in the paper fail to update permissions promptly.
Similar Scheme Functional Comparison
|
Scheme |
Signcryptor Privacy |
Multi-Agency |
Verifiability |
Audit |
Outsourcing calculation |
IPFS |
|
[63] |
× |
× |
× |
× |
× |
× |
|
[25] |
√ |
× |
√ |
× |
√ |
× |
|
[8] |
√ |
× |
× |
× |
× |
× |
|
[49] |
√ |
√ |
√ |
√ |
√ |
× |
|
ours |
√ |
√ |
√ |
√ |
√ |
√ |
Where “√” indicates that the feature exists in the scheme, and “×” means that the feature does not exist.
Through these specific revisions and additions, we aim to enhance the innovation and academic value of the paper, aligning it more closely with the expectations of the reviewer and editor. We will also continue to address the reviewer’s other suggestions to ensure the paper meets the highest quality standards in all aspects.
Once again, thank you for your valuable time and professional review. We look forward to this round of thorough revisions significantly improving the academic level of our paper and contributing a valuable piece of literature to the relevant field.

Reviewer 3 Report
Comments and Suggestions for Authors
The authors have addressed most of my comments. Here are some minor comments.
1. In the introduction, the authors does not make clear the significance of this work. They achieved data sharing based on attribute signcryption and blockchain. However, there are many similar schemes, and the authors did not discuss the shortcomings of these schemes, which makes it difficult to understand the significance of the proposed work.
2. The authors need to further highlight their contributions. In contribution a), they mentioned “We introduce an innovative approach for a data security sharing paradigm that utilizes blockchain and attribute-based signcryption.” However, there are many data sharing solutions based on blockchain and attribute-based signcryption. Compared with these solutions, this contribution does not seem to be novel enough.
3. This authors need to introduce more cutting-edge references. In Section 2.1, the data storage sharing schemes [28], [29] cited by the authors are relatively cutting-edge. However, the references on ABSC are not the latest publications. It is recommended to add references in 2024 and analyze the advantages and disadvantages of similar schemes.
Y. S. Rao, S. Prasad, S. Bera, A. K. Das and W. Susilo, "Boolean Searchable Attribute-Based Signcryption With Search Results Self-Verifiability Mechanism for Data Storage and Retrieval in Clouds," in IEEE Transactions on Services Computing, vol. 17, no. 4, pp. 1382-1399, July-Aug. 2024.
Vulnerability of Machine Learning Approaches Applied in IoT-based Smart Grid: A Review, IEEE Internet of Things Journal, 2024, vol. 11, no. 11, pp. 18951-18975, June 1, 2024.
Author Response
Comments 1: In the introduction, the authors does not make clear the significance of this work. They achieved data sharing based on attribute signcryption and blockchain. However, there are many similar schemes, and the authors did not discuss the shortcomings of these schemes, which makes it difficult to understand the significance of the proposed work.
Response 1: Thank you for the valuable comments from the reviewer. Indeed, clearly articulating the significance of the work in the introduction is crucial, as it helps readers understand the motivation and innovations of the research. We have revised the introduction based on the reviewer's suggestions, particularly in lines 82 to 129, where we have added discussions on the shortcomings of existing similar solutions and emphasized the advantages and significance of our approach. Below is the revised introduction:
Wu et al. [14] designed a system model that includes data users, IPFS, and blockchain, proposing the storage of personal health records on IPFS. Although the overall design is innovative and secure, it still presents many security risks and vulnerabilities. First, the lack of a detailed mechanism for fine-grained access control increases the risk of data leakage to unauthorized users. Second, the identity authentication mechanism is not clearly defined, and metadata is stored on the blockchain without encryption, which can lead to privacy breaches and data integrity issues. Azbeg et al. [15] proposed a healthcare data security storage solution based on IPFS and blockchain, where data is encrypted using proxy re-encryption. However, the use of a single smart contract for access control can cause response delays in high-concurrency scenarios. Similarly, Rani et al. [16] introduced a remote patient monitoring scheme based on blockchain, utilizing IPFS for data storage and sharing and DApps for data collection and connection to the blockchain. This solution also faces the limitation of relying on a single smart contract for access control. Jayabalan and Jeyanthi [17] proposed an electronic health record management system combining blockchain and IPFS. Although the framework incorporates multiple security mechanisms, it still faces some risk challenges. Specifically, in a patient-centered access model, there are issues with key management, and if attackers intercept digital envelopes, they may attempt to crack the symmetric keys, leading to data decryption and privacy leaks.
In the field of data sharing, solutions based on attribute-based encryption and blockchain technology have been extensively researched and applied. However, despite significant progress in enhancing data security, existing data sharing schemes still have several limitations. For instance, some schemes rely on complex key management mechanisms, which not only increase system overhead but also introduce single points of failure, threatening overall security. Additionally, some schemes have coarse-grained access control, lacking fine-grained permission management or offering limited and inflexible access control methods. Furthermore, existing solutions often overlook privacy protection during the data sharing process, leading to potential leaks of sensitive information.
To address these issues, we propose a new paradigm for secure data sharing. This paradigm integrates attribute-based signcryption with blockchain technology. By employing attribute certification mechanisms and smart contract-driven access control, it achieves more granular and secure access control and data sharing. Moreover, the immutability and transparency of blockchain ensure data integrity and traceability, while the decentralized storage of IPFS improves data access efficiency and resistance to censorship. Additionally, our solution places a strong emphasis on data privacy protection, using signcryption techniques to process data and effectively prevent the leakage of sensitive information. Furthermore, the inclusion of auditors allows for the tracing and review of transaction records on the blockchain, enabling the identification and correction of errors and anomalies, thus ensuring the accuracy and compliance of the data sharing process.
We believe that these modifications will enable readers to better appreciate the significance and value of our work. We also look forward to further feedback and suggestions from the reviewer on the revised introduction. Thank you for your time and effort.
Comments 2: The authors need to further highlight their contributions. In contribution a), they mentioned “We introduce an innovative approach for a data security sharing paradigm that utilizes blockchain and attribute-based signcryption.” However, there are many data sharing solutions based on blockchain and attribute-based signcryption. Compared with these solutions, this contribution does not seem to be novel enough.
Response 2: Thank you for the valuable comments from the reviewers. We have reorganized and deepened the section on the contributions of our paper. Regarding contribution a), we realized that the original description might not have sufficiently highlighted the uniqueness and innovativeness of our work. Therefore, we have revised this part to more explicitly showcase the distinctions between our contributions and existing solutions. The revised contributions are as follows:
- By proposing a dual authentication mechanism based on smart contracts and trusted authorities, and integrating a multi-attribute-based access control strategy, a more flexible, efficient, and verifiable access control approach has been jointly developed.
- A data privacy protection mechanism based on attribute-based signcryption has been established, simplifying key management, reducing overhead, and enhancing the security of data sharing.
We hope this revision can more accurately convey the core value and uniqueness of our work. We look forward to receiving further valuable guidance and suggestions from the reviewers. Once again, we are grateful for the reviewers’ patient review and professional insights.
Comments 3: This authors need to introduce more cutting-edge references. In Section 2.1, the data storage sharing schemes [28], [29] cited by the authors are relatively cutting-edge. However, the references on ABSC are not the latest publications. It is recommended to add references in 2024 and analyze the advantages and disadvantages of similar schemes.
- S. Rao, S. Prasad, S. Bera, A. K. Das and W. Susilo, "Boolean Searchable Attribute-Based Signcryption With Search Results Self-Verifiability Mechanism for Data Storage and Retrieval in Clouds," in IEEE Transactions on Services Computing, vol. 17, no. 4, pp. 1382-1399, July-Aug. 2024.
Vulnerability of Machine Learning Approaches Applied in IoT-based Smart Grid: A Review, IEEE Internet of Things Journal, 2024, vol. 11, no. 11, pp. 18951-18975, June 1, 2024.
Response 3: Thank you for your valuable suggestions. We greatly appreciate your detailed review of the paper. Regarding the issue of updating the references in Section 2.1, we fully agree and will focus on improving this in the subsequent revisions. Specifically, we plan to add the latest ABSC-related literature from 2024, which will cover the most recent research trends and technological breakthroughs in the field. While adding these references, we will also conduct a thorough analysis of the strengths and weaknesses of these similar solutions, highlighting the innovative and practical value of our approach. The reviewer mentioned that the data storage and sharing schemes cited by the authors [28], [29] are quite advanced. Since Section 2.2 of the paper discusses data sharing schemes, we have moved references [28] and [29] to this section and adjusted their order accordingly. The additional content is as follows:
Rao et al. [28] introduced a secure searchable attribute-based signcryption scheme, which supports efficient search over signed-and-encrypted data, keyword privacy protection, and self-verification of search results. Despite its contributions, the scheme’s complexity, performance overhead, and security concerns regarding cloud servers cannot be overlooked. For example, the verification process may be overly complex for ordinary users, increasing the likelihood of operational errors. Additionally, computationally intensive tasks are executed locally or on devices with limited resources rather than being outsourced, leading to increased response times and inadaptability to high-concurrency scenarios. Furthermore, cloud servers log users’ search behaviors and results, and any leakage of these logs could compromise user privacy. Zhang et al. [29] presented a comprehensive review of machine learning applications in smart grids, with a particular focus on the security implications of power system characteristics in machine learning-based smart grid applications (MLsgAPP). This review fills a significant research gap in this area, particularly addressing the issues of security and adversarial attacks in MLsgAPP, while providing practical application insights and future research directions. However, the discussion lacks detailed exploration of mathematical models or implementation specifics, limiting readers’ understanding of certain technical details. Moreover, the dynamic nature of smart grids necessitates access control mechanisms that can rapidly adapt to changes, yet the access control methods mentioned in the paper fail to update permissions promptly.
Gao et al. [40] designed a decentralized storage scheme based on blockchain and IPFS. This scheme employs ciphertext-policy attribute-based encryption (CP-ABE) to encrypt symmetric keys, supporting fine-grained access control where the granularity is based on attribute-based access control policies. Zhang et al. [41] developed a chameleon signcryption mechanism and designed a ciphertext authentication protocol, which excels in ensuring identity verification between avatars and the public verifiability of encrypted identities, making it suitable for metaverse scenarios. However, it requires enhancement in the applicability of access control mechanisms and the establishment of comprehensive auditing and monitoring systems to promptly detect and respond to anomalous behaviors, thereby preventing potential attacks.
We hope that these efforts will better demonstrate the technical depth and innovative value of our research, providing more comprehensive and detailed references for peer review. Additionally, we will continue to monitor the latest developments in the field to ensure that our work remains at the forefront of academic research.
Once again, thank you for your valuable time and professional feedback. We look forward to significantly improving the quality of the paper through this round of careful revisions, and contributing a valuable outcome to the academic community.
